# Meta 3D AssetGen: Text-to-Mesh Generation with High-Quality Geometry, Texture, and PBR Materials

Yawar Siddiqui[†]    Tom Monnier*    Filippos Kokkinos*    Mahendra Kariya
Yanir Kleiman    Emilien Garreau    Oran Gafni    Natalia Neverova
Andrea Vedaldi    Roman Shapovalov*    David Novotny*

GenAI, Meta        [†]TU Munich; intern with Meta    *core technical contributors

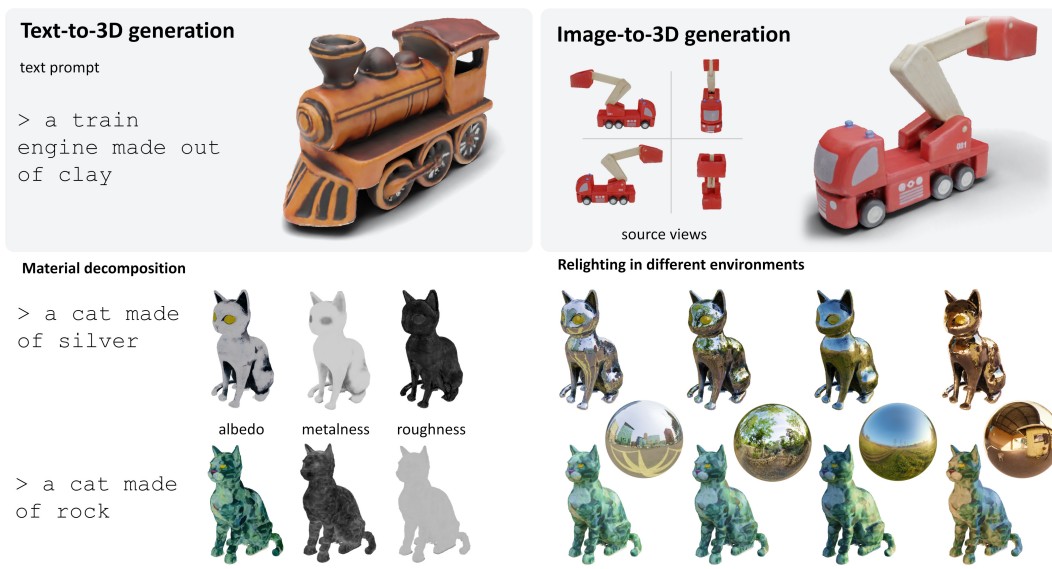

Figure 1: We present **Meta 3D AssetGen**, a novel text- or image-conditioned generator of 3D meshes with physically-based rendering materials (top). Meta 3D AssetGen produces meshes with detailed geometry and high-quality textures, and decomposes materials into albedo, metalness, and roughness (bottom left), which allows to realistically relight objects in new environments (bottom right).

## Abstract

We present Meta 3D AssetGen (AssetGen), a significant advancement in text-to-3D generation which produces faithful, high-quality meshes with texture and material control. Compared to works that bake shading in the 3D object's appearance, AssetGen outputs physically-based rendering (PBR) materials, supporting realistic relighting. AssetGen generates first several views of the object with separate shaded and albedo appearance channels, and then reconstructs colours, metalness and roughness in 3D, using a deferred shading loss for efficient supervision. It also uses a sign-distance function to represent 3D shape more reliably and introduces a corresponding loss for direct shape supervision. This is implemented using fused kernels for high memory efficiency. After mesh extraction, a texture refinement transformer operating in UV space significantly improves sharpness and details. AssetGen achieves 17% improvement in Chamfer Distance and 40% in LPIPS over the best concurrent work for few-view reconstruction, and a human preference of 72% over the best industry competitors of comparable speed, including those that support PBR. Project page with generated assets: `https://assetgen.github.io`

38th Conference on Neural Information Processing Systems (NeurIPS 2024).

# 1 Introduction

Generating 3D objects from text or image prompts has enormous potential for 3D graphics, with applications in animation, gaming and virtual reality. However, while image and video generators have improved dramatically [68, 44, 55, 42, 97, 103], 3D generators are not ready yet for professional use. In fact, 3D generators are often slow and produce artifacts in the generated 3D meshes and textures. Many 3D generators, furthermore, "bake" appearance as albedo, ignoring how materials respond to variable environmental illumination. This results in visually unattractive outputs, especially for reflective materials, which look out of place when put in novel environments.

In this paper, we introduce *Meta 3D AssetGen*, a significant step-up in text-conditioned 3D generation. AssetGen generates assets in under 30 seconds while outperforming prior methods of comparable speed in faithfulness, in quality of the generated 3D meshes and, especially, in quality and control of materials, by supporting *Physically-Based Rendering* (PBR) [87]. The model generates albedo, metalness, and roughness so that rendered scenes can accurately reflect environmental illumination. In addition, we focus on meshes as the output representation due to their prevalence in applications and compatibility with PBR.

AssetGen uses the two-stage design epitomized by [42]. The first stage *stochastically* generates four images of the object from four canonical viewpoints, and the second stage *deterministically* reconstructs the 3D shape, appearance and materials of the object from these views (Fig. 1). The two-stage approach is faster and more robust than SDS-based techniques that perform test-time optimization [68] and, so far, produces better results than single-stage 3D generators [37, 61, 94, 79].

The first question we ask is how this design should be extended to support PBR. We show that it is difficult for the image-to-3D stage to predict PBR channels from an image as this problem is ambiguous and the model is deterministic. However, we also show that it is difficult offload PBR prediction to the text-to-image model; while this is stochastic, which handles ambiguity, the PBR channels are statistically different from the natural images used for pre-training, which makes fine-tuning difficult. Our solution is to give the text-to-image model the simpler task of outputting shaded appearance and albedo only, and task the image-to-3D stage with inferring the PBR channels from these. This reduces the statistical gap for the text-to-image model and still removes most of the ambiguity for the image-to-3D model.

We also note that the quality of 3D shapes and meshes is crucial for PBR modelling. Hence, the second question we study is how to improve 3D quality. We do so by learning a reconstruction network, *MetaILRM*, which outputs directly a signed-distance field (SDF). SDFs are better than opacity fields for meshing, as the zero level set of an SDF traces the object's surface more reliably. Furthermore, the SDF can be directly supervised using ground-truth depth maps, which is not immediately possible for opacity. The crucial contribution here is to add SDF support, including the VolSDF [108] formulation for differentiable rendering, to the memory-efficient Lightplane kernels [6]. In this way, we can use the stronger SDF representation together with larger batches and photometric loss supervision on high-resolution renders, improving both shapes and textures.

Finally, we note that much of the quality of the final asset depends on texture quality. MetaILRM's textures can still be slightly blurrier than the input image due to the limited resolution of the volumetric representation. The third question we investigate is how to maximize the texture quality. To this end, we introduce a new texture refiner network which upgrades the extracted albedo and materials by fusing information extracted from the original views, resolving possible conflicts between them.

We demonstrate the effectiveness of AssetGen on the image-to-3D and text-to-3D tasks. For image-to-3D, we attain state-of-the-art performances among existing few-view mesh-reconstruction methods when measuring the accuracy of the recover shaded and PBR texture maps. For text-to-3D, we conduct extensive user studies to compare the best methods from academia and industry that have comparable inference time, and outperform them in terms of visual quality and text alignment.

# 2 Related Work

**Text-to-3D.** Inspired by text-to-image models, early text-to-3D approaches [39, 31, 64, 34, 26, 110] train 3D diffusion models on datasets of captioned 3D assets. Yet, the limited size and diversity of 3D data prevents generalization to open-vocabulary prompts. Recent works thus pivoted into basing

such generators on text-to-image models that are trained on billions of captioned images. Among these, works like [75, 56] finetune 2D diffusion models to output 3D representations, but the quality is limited due to the large 2D-3D domain gap. Other approaches can be dived into two groups.

The first group contains methods that build on DreamFusion, a seminal work by [68], and distill 3D objects by optimizing NeRF via the SDS loss, matching its renders to the belief of a pre-trained text-to-image model. Extensions have considered: (i) other 3D representations like hash grids [44, 69], meshes [44] and 3D Gaussians (3DGS) [81, 111, 12]; (ii) improved SDS [91, 95, 119, 30]; (iii) monocular conditioning [69, 82, 113, 78]; (iv) predicting additional normals or depth for better geometry [70, 78]. Yet, distillation methods are prone to issues such as the Janus effect (duplicating object parts) and content drift [74]. A common solution is to incorporate view-consistency priors into the diffusion model, by either conditioning on cameras [47, 73, 32, 11, 69] or by generating multiple object views jointly [74, 98, 93, 40, 118]. Additionally, SDS optimization is slow and requires minutes to hours per assets; this issue is partly addressed in [52, 101] with amortized SDS.

The second group of methods includes faster two-stage approaches [46, 51, 49, 107, 106, 8, 83, 28, 23] that start by generating multiple views of the object using a text-to-image or text-to-video model [54, 13] tuned to output multiple views of the object followed by per-scene optimization using NeRF [58] or 3DGS [38]. However, per-scene optimization requires several highly-consistent views which are difficult to generate reliably. Instant3D [42] improves speed and robustness by generating a grid of just four views followed by a feed-forward network (LRM [29]) that reconstructs the object from these. One-2–3–45++ [45] replaces the LRM with a 3D diffusion model. Our AssetGen builds on the Instant3D paradigm and upgrades the LRM to output PBR materials and an to use a SDF-based representation of 3D shape. Furthermore, it starts from grids of four views with shaded and albedo channels, key to predicting accurate 3D shape and materials from images.

**3D reconstruction from images.** 3D scene reconstruction, in its traditional *multi-view stereo* (MVS) sense, assumes access to a dense set of scene views. Recent reconstruction methods such as NeRF [58] optimize a 3D representation by minimizing multi-view rendering losses. There are two popular classes of 3D representation: (i) explicit representations like meshes [22, 114, 24, 63, 59, 76] or 3D points/Gaussians [38, 25], and (ii) implicit representations like occupancy fields [65], radiance fields [58, 62] and signed distance functions (SDF) [109]. Compared to occupancy fields, SDF [66, 108, 92, 17, 21] simplifies surface constraints integration, improving scene geometry. For this reason, we also use SDFs and demonstrate that they outperform occupancy fields.

*Sparse-view reconstruction* instead assumes few input views (usually 1 to 8). An approach to mitigate the lack of dense multiple views is to leverage 2D diffusion priors in optimization [55, 99], but this is often slow and fragile. More recently, authors have focused on training feed-forward reconstructors on large datasets [14, 36, 57, 48, 100, 60, 94]. In particular, [29] trains a large Transformer [89] to predict NeRF using a triplane representation [7, 9]. Followups study 3D representations like meshes [103, 97] and 3DGS [120, 105, 80, 115], improved backbones [96, 97] and training protocols [86, 35]. We introduce three extensions to LRM: (i) an SDF formulation for improved geometry, (ii) PBR material prediction for relighting, and (iii) a texture refiner for better texture details.

**3D modeling with PBR materials.** Most 3D generators output 3D objects with baked illumination, either view-dependent [58, 38] or view-independent [29]. Since baked lighting ignores the model's response to environmental illumination, it is unsuitable for graphics pipelines that simulate lighting. Physically-based rendering (PBR) defines material properties so that a suitable shader can account for illumination realistically. Several MVS works have considered estimating PBR materials using NeRF [4, 3, 102], SDF [116], differentiable meshes [63, 27] or 3DGS [33, 43]. In generative modelling, [10, 70, 50, 104] augment the text-to-3D SDS optimization [68] with a PBR model. Differently from them, we integrate PBR modeling in our feed-forward text-to-3D network, unlocking for the first time fast text-based generation of 3D assets with controllable PBR materials.

## 3 Method

AssetGen is a two-stage pipeline (Fig. 2). The first stage, text-to-image (Sec. 3.1), maps text to an image grid containing four object views with material information. The second stage, image-to-3D, comprises a novel PBR-based sparse-view reconstruction model (Sec. 3.2) and a new texture refiner (Sec. 3.3). As such, AssetGen is applicable to two tasks: text-to-3D (stage 1+2) and image-to-3D (stage 2 only).

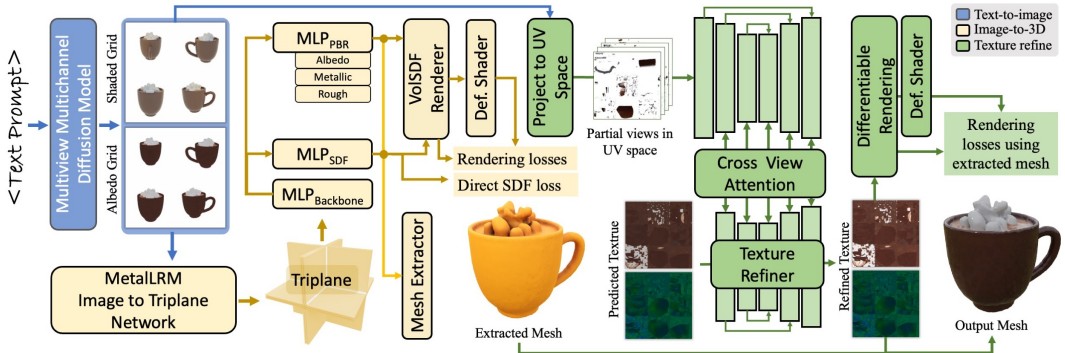

Figure 2: **Overview.** Given a text prompt, AssetGen generates a 3D mesh with PBR materials in two stages. The first text-to-image stage (blue) predicts a 6-channel image depicting 4 views of the object with shaded and albedo colors. The second image-to-3D stage includes two steps. First, a 3D reconstructor (dubbed MetaILRM) outputs a triplane-supported SDF field converted into a mesh with textured PBR materials (orange). Then, PBR materials are enhanced with our texture refiner which recovers missing details from the input views (green).

## 3.1 Text-to-image: Generating shaded and albedo images from text

The goal of the text-to-image module is to generate several views of the generated 3D object. To this end, we employ an internal text-to-image diffusion model pre-trained on billions of text-annotated images, with an architecture similar to Emu [16]. Similar to [74, 42], we finetune the model to predict a grid of four images $I_i$, $i = 1, \ldots, 4$, each depicting the object from canonical viewpoints $\pi_i$. Note that $I_i$ are RGB images of the shaded object. We tried deferring the PBR parameter extraction to the image-to-3D stage, but this led to suboptimal results. This is due to the determinism of the image-to-3D stage, which fails to model ambiguities when assigning materials to surfaces.

A natural solution, then, is to predict the PBR parameters directly in the text-to-image stage. These consists of the albedo $\rho_0$ (by which we mean the base color, which is the same as albedo only for zero metalness), the metalness $\gamma$, and the roughness $\alpha$. However, we found this to be ineffective too because the metalness and roughness maps deviate from the distribution of natural images making them a hard target for finetuning. Our novel solution is to train the model to generate instead a **4-view grid with 6 channels**, 3 for the shaded appearance $I$ and 3 more for the albedo $\rho_0$. This reduces the finetuning gap, and removes enough ambiguity for accurate PBR prediction in the image-to-3D stage.

## 3.2 Image-to-3D: A PBR-based large reconstruction model

We now describe the image-to-3D stage, which solves the reconstruction tasks given either a small number of views $I_i$ (few-view reconstruction), or the 4-view 6-channel grid of Sec. 3.1.

At the core of our method is a new PBR-aware reconstruction model, MetaILRM, that reconstructs the object given $N \geq 1$ *posed* images $(I_i, \pi_i)_{i=1}^N$, where $I_i \in \mathbb{R}^{H \times W \times D}$ and $\pi_i \in \Pi$ is the camera viewpoint. As noted in Sec. 3.1, we consider $N = 4$ canonical viewpoints $\pi_1, \ldots, \pi_4$ (fixed to 20° elevation and 0°, 90°, 180°, 270° azimuths) and $D = 6$ input channels. The output is a 3D field representing the shape and PBR materials of the object as an SDF $s : \mathbb{R}^3 \to \mathbb{R}$, where $s(\boldsymbol{x})$ is the signed distance from the 3D point $\boldsymbol{x}$ to the nearest object surface point, and a PBR function $k : \mathbb{R}^3 \to \mathbb{R}^5$, where $k(\boldsymbol{x}) = (\rho_0, \gamma, \alpha)$ are the albedo, metalness and roughness.

The key to learning the model is the *differentiable rendering* operator $\mathcal{R}$. This takes as input a field $\ell : \mathbb{R}^3 \to \mathbb{R}^D$, the SDF $s$, the viewpoint $\pi$, and a pixel $u \in U = [0, W] \times [0, H]$, and outputs the projection of the field on the pixel according to the rendering equation [58], which has the same number of channels $D$ as the rendered field $\ell$:

$$\mathcal{R}(u \mid \ell, s, \pi) = \int_0^\infty \ell(\boldsymbol{x}_t) \sigma(\boldsymbol{x}_t \mid s) e^{- \int_0^t \sigma(\boldsymbol{x}_\tau \mid s) \, d\tau} dt. \qquad (1)$$

Here $\boldsymbol{x}_t = \boldsymbol{x}_0 - t\boldsymbol{\omega}_o$, $t \in [0, \infty)$ is the ray that goes from the camera center $\boldsymbol{x}_0$ through the pixel $u$ along direction $-\boldsymbol{\omega}_o \in \mathbb{S}^2$. The function $\sigma(\boldsymbol{x} \mid s)$ is the opacity of the 3D point $\boldsymbol{x}$ and is obtained

from the SDF value $s(\boldsymbol{x})$ using the VolSDF [108] formula

$$\sigma(\boldsymbol{x} \mid s) = \frac{a}{2}\Big(1 + \operatorname{sign} s(\boldsymbol{x})\big(1 - e^{-|s(\boldsymbol{x})|/b}\big)\Big), \tag{2}$$

where $a, b$ are the hyper-parameters. We use Eq. (1) to render several different types of fields $\ell$, the most important of which is the radiance field, introduced next along with the material model.

**Reflectance model.** The appearance of the object $\hat{I}(u) = \mathcal{R}(u \mid L, s, \pi)$ in a shaded RGB image $\hat{I}$ is obtained by rendering its *radiance field* $\ell(\boldsymbol{x}) = L(\boldsymbol{x}, \boldsymbol{\omega}_{\mathrm{o}} \mid k, \boldsymbol{n})$, where $\boldsymbol{n}$ is the field of unit normals. The radiance is the light reflected by the object in the direction $\boldsymbol{\omega}_{\mathrm{o}}$ of the observer (see App. A.8 for details), which in PBR is given by:

$$L(\boldsymbol{x}, \boldsymbol{\omega}_{\mathrm{o}} \mid k, \boldsymbol{n}) = \int_{H(\boldsymbol{n})} f(\boldsymbol{\omega}_{\mathrm{i}}, \boldsymbol{\omega}_{\mathrm{o}} \mid k(\boldsymbol{x}), \boldsymbol{n}(\boldsymbol{x})) L(\boldsymbol{x}, -\boldsymbol{\omega}_{\mathrm{i}})(\boldsymbol{n}(\boldsymbol{x}) \cdot \boldsymbol{\omega}_{\mathrm{i}}) \, d\Omega_{\mathrm{i}}, \tag{3}$$

where $\boldsymbol{\omega}_{\mathrm{o}}, \boldsymbol{\omega}_{\mathrm{i}} \in H(\boldsymbol{n}) = \{\boldsymbol{\omega} \in \mathbb{S}^2 : \boldsymbol{n} \cdot \boldsymbol{\omega} \geq 0\}$ are two unit vectors pointing outside the object and $L(\boldsymbol{x}, -\boldsymbol{\omega}_{\mathrm{i}})$ is the radiance incoming from the environment at $\boldsymbol{x}$ from direction $\boldsymbol{\omega}_{\mathrm{i}}$ in the solid angle $d\Omega_{\mathrm{i}}$. The *Bidirectional Reflectance Distribution Function* (BRDF) $f$ tells how light received from direction $-\boldsymbol{\omega}_{\mathrm{i}}$ (incoming) is scattered into different directions $\boldsymbol{\omega}_{\mathrm{o}}$ (outgoing) by the object [20].

In PBR, we consider a physically-inspired model for the BRDF, striking a balance between realism and complexity [2, 87, 77, 15, 90]; specifically, we use the Disney GGX model [90, 5], which depends on parameters $\rho_0$, $\gamma$, and $\alpha$ only (see App. A.12.1 for the parametric form of $f$). Hence, the MetaI LRM predicts the triplet $k(\boldsymbol{x}) = (\rho_0, \gamma, \alpha)$ at each 3D point $\boldsymbol{x}$.

**Deferred shading.** In practice, instead of computing $\hat{I}(u) = \mathcal{R}(u \mid L, s, \pi)$ using Eqs. (1) and (3), we use the process of *deferred shading* [20]:

$$\hat{I}(u) = \mathcal{R}_{\mathrm{def}}(u \mid k, s, \pi) = \int_{H(\boldsymbol{n})} f(\boldsymbol{\omega}_{\mathrm{i}}, \boldsymbol{\omega}_{\mathrm{o}} \mid \bar{k}, \bar{\boldsymbol{n}}) L_{\mathrm{env}}(-\boldsymbol{\omega}_{\mathrm{i}})(\bar{\boldsymbol{n}} \cdot \boldsymbol{\omega}_{\mathrm{i}}) \, d\Omega_{\mathrm{i}}, \tag{4}$$

where $L_{\mathrm{env}}$ is the environment radiance (assumed to be the same for all $\boldsymbol{x}$), $\bar{k} = \mathcal{R}(u \mid k, s, \pi)$ and $\bar{\boldsymbol{n}} = \mathcal{R}(u \mid \boldsymbol{n}, s, \pi)$ are rendered versions of the material and normal fields. The advantage of Eq. (4) is that the BRDF $f$ is evaluated only once per pixel, which is much faster and less memory intensive than doing so for each 3D point during the evaluation of Eq. (1), particularly for training/backpropagation. During training, furthermore, the environment light is assumed to be a single light source at infinity, so the integral (4) reduces to evaluating a single term.

**Training formulation and losses.** MetaILRM is thus a neural network that takes as input a set of images $(I_i, \pi_i)_{i=1}^N$ and outputs estimates $\hat{s}$ and $\hat{k}$ for the SDF and PBR fields. We train it from a dataset of mesh surfaces $M \subset \mathbb{R}^3$ with ground truth PBR materials $k : M \to \mathbb{R}^5$.

Reconstruction models are typically trained via supervision on renders [29, 97]. However, physically accurate rendering via Eq. (1) is very expensive. We overcome this hurdle in two ways. First, we render the raw ground-truth PBR fields $k$ and use them to supervise their predicted counterparts with the MSE loss, skipping Eq. (1). For the rendered albedo $\rho_0$ — which is similar enough to natural images — we also use the LPIPS [117] loss:

$$\mathcal{L}_{\mathrm{pbr}} = \mathrm{LPIPS}\Big(\mathcal{R}(\cdot \mid \hat{\rho}_0, \hat{s}, \pi), \mathcal{R}(\cdot \mid \rho_0, M, \pi)\Big) + \Big\|\mathcal{R}(\cdot \mid \hat{k}, \hat{s}, \pi) - \mathcal{R}(\cdot \mid k, M, \pi)\Big\|^2. \tag{5}$$

We further supervise the PBR field by adding a computationally-efficient deferred shading loss:

$$\mathcal{L}_{\mathrm{def}} = \|\sqrt{w} \odot (\mathcal{R}_{\mathrm{def}}(\cdot \mid \hat{k}, \hat{s}, \pi) - \mathcal{R}_{\mathrm{def}}(\cdot \mid k, M, \pi))\|^2. \tag{6}$$

The weight $w(u) = \hat{\boldsymbol{n}}(u) \cdot \boldsymbol{n}(u)$ is the dot product of the predicted and ground-truth normals at pixel $u$. It discounts the loss where the predicted geometry is not yet learnt. Fig. 14 (b) visualizes deferred shading and the rendering loss.

Finally, we also supervise the SDF field with a direct loss $\mathcal{L}_{\mathrm{sdf}}$ (implemented as in [1]), a depth-MSE loss $\mathcal{L}_{\mathrm{depth}}$ between the depth renders and the ground truth, and with a binary cross-entropy $\mathcal{L}_{\mathrm{mask}}$ between the alpha-mask renders and the ground-truth masks. Refer to App. A.6.2 for more details.

**LightPlane implementation.** We base MetaILRM on LightplaneLRM [6], a variant of LRM [29] exploiting memory and compute-efficient Lightplane splatting and rendering kernels, offering better quality reconstructions. However, since LightplaneLRM uses density fields, which are suboptimal for mesh conversion [92, 66, 1], we extend the Lightplane rendering GPU kernel with a VolSDF [108] renderer using Eq. (2). Additionally, we also fuse into the kernel the direct SDF loss $\mathcal{L}_{\mathrm{sdf}}$ since a naive autograd implementation is too memory-heavy.

### 3.3  Mesh extraction and texture refiner

The MetaILRM module of Sec. 3.2 outputs a sign distance function $s$, implicitly defining the object surface $A = \{\boldsymbol{x} \in \mathbb{R}^3 \mid s(\boldsymbol{x}) = 0\}$ as a level set of $s$. We use the Marching Tetrahedra algorithm [18] to trace the level set and output a mesh $M \approx A$. Then, xAtlas [112] extracts a UV map $\phi : [0, V]^2 \to M$, mapping each 2D UV-space point $v = \phi(\boldsymbol{x})$ to a point $\boldsymbol{x} \in M$ on the mesh.

Next, the goal is to extract a high-quality 5-channel PBR texture image $\bar{K} \in \mathbb{R}^{V \times V \times 5}$ capturing the albedo, metalness, and roughness of each mesh point. The texture image $K$ can be defined directly by sampling the predicted PBR field $\hat{k}$ as $K(v) \leftarrow \hat{k}(\phi(v))$, but this often yields blurry results due to the limited resolution of MetaILRM. Instead, we design a texture refiner module which takes as input the coarse PBR-sampled texture image as well as the $N$ views representing the object and outputs a much sharper texture $\bar{K}$. In essence, this modules leverages the information from the different views to refine the coarse texture image. The right part of Fig. 2 illustrates this module.

More specifically, it relies on a network $\Phi$ which is fed $N + 1$ texture images $\{K_i\}_{i=0}^{N}$. First, each pixel $v \in [0, V]^2$ of $K_0 \in \mathbb{R}^{V \times V \times 11}$ is annotated with the concatenation of the normal, the 3D location, and the output of MetaILRM's PBR field $k(\phi(v))$ evaluated at $v$'s 3D point $\phi(v)$. The remaining $K_1, \ldots, K_N$ correspond to partial texture images with 6 channels (for the base and shaded colors) which are obtained by back-projecting the object views to the mesh surface. The network $\Phi$ utilises two U-Nets to fuse $\{K_i\}_{i=0}^{N}$ into the enhanced texture $\bar{K}$. $\Phi$'s goal is to select, for each UV point $v$, which of the $N$ input views provides the best information. Specifically, each partial texture image $K_i$ is processed in parallel by a first U-Net, and the resulting information is communicated via cross attention to a second U-Net whose goal is to refine $K_0$ into the enhanced texture $\bar{K}$. Please refer to App. A.7 for further details.

Such a network is trained on the same dataset and supervised with the PBR and albedo rendering losses as MetaILRM. The only difference is meshes (whose geometry is fixed) are rendered differentiably using PyTorch3D's [71] mesh rasterizer instead of the Lightplane SDF renderer.

## 4  Experiments

Our **training data** consists of 140,000 meshes of diverse semantic categories created by 3D artists. For each asset, we render 36 views at random elevations within the range of $[-30°, 50°]$ at uniform intervals of $30°$ around the object, lit with a randomly selected environment map. We render the shaded images, albedo, metalness, roughness, depth maps, and foreground masks from each viewpoint. The text-to-image stage is based on an internal text-to-image model architecturally similar to Emu [16], fine-tuned on a subset of 10,000 high-quality 3D samples, captioned by a Cap3D-like pipeline [53] that uses Llama3 [88]. The other stage utilizes the entire 3D dataset instead.

For **evaluation**, following [105, 103, 42], we assess visual quality using PSNR and LPIPS [117] between the rendered and ground-truth images. PSNR is computed in the foreground region to avoid metric inflation due to the empty background. Geometric quality is measured by the L1 error between the rendered and ground-truth depth maps (of the foreground pixels), as well as the IoU of the object silhouette. We further report Chamfer Distance (CD) and Normal Correctness (NC) for 20,000 points uniformly sampled on both the predicted and ground-truth shapes. Material decomposition

Table 1: **Four-view reconstruction with PBR** evaluating the accuracy of the PBR renders for MetaI LRM and ablations. Methods in top / bottom accept 4 views with shaded / shaded&albedo color channels.

| Method | LPIPS↓ albedo | PSNR↑ albedo | metal | rough |
|---|---|---|---|---|
| C = LightplaneLRM w/ SDF | 0.117 | 17.14 | 12.39 | 15.25 |
| E = C + Material prediction | 0.097 | 20.66 | 15.99 | 20.25 |
| F = E + Deferred shading loss | 0.093 | 21.12 | 18.64 | 20.66 |
| G = F + Texture refinement | **0.087** | **21.97** | **22.19** | **20.85** |
| H = F + Albedo & shaded input | 0.084 | 23.02 | 20.43 | **21.18** |
| I = H + Texture refinement | **0.069** | **24.39** | **27.28** | 20.63 |

Table 2: **Win-rate of AssetGen in text-to-3D user study** evaluating visual quality and the alignment between the prompt and the generated meshes. AssetGen beats all baselines at 30 sec budget (on an A100 GPU).

| Method | Visual quality | Text fidelity | PBR |
|---|---|---|---|
| GRM [105] | 96.7 % | 93.3 % | ✗ |
| InstantMesh [103] | 99.3 % | 97.3 % | ✗ |
| LightplaneLRM [6] | 66.6 % | N/A | ✗ |
| Meshy v3 [85] | 94.6 % | 91.3 % | ✓ |
| Luma Genie 1.0 [84] | 72.3 % | 72.8 % | ✓ |

is evaluated with LPIPS and PSNR on the albedo image, and PSNR alone for the metalness and roughness channels. All metrics are calculated on meshified outputs rather than on neural renders.

## 4.1 Sparse-view reconstruction

We tackle the sparse-view reconstruction task of predicting a 3D mesh from 4 posed images of an object on a subset of 332 meshes from Google Scanned Objects (GSO) [19]. We compare against state-of-the-art Instant3D-LRM [42], GRM [105], InstantMesh [103], and MeshLRM [97]. We also include LightplaneLRM [6], an improved version of Instant3D-LRM, which serves as our base model. MeshLRM [97] has not been open-sourced so we compare only qualitatively to meshes from their webpage. All methods are evaluated using the

Table 3: **Four-view reconstruction on GSO** comparing the appearance and geometry of MetalLRM (outputting baked-light texture) to baselines (top) and ablations (bottom). CD values multiplied by $10^{-2}$.

| Method | LPIPS↓ | PSNR↑ | Depth↓ | IoU↑ | CD↓ | NC↑ |
|---|---|---|---|---|---|---|
| Instant3D-LRM [29] | 0.124 | 18.54 | 0.325 | 0.930 | 1.630 | 0.844 |
| GRM [105] | 0.100 | 19.87 | 0.364 | 0.949 | 1.490 | 0.873 |
| InstantMesh [103] | 0.113 | 20.63 | 0.334 | 0.937 | 1.364 | 0.848 |
| **MetalLRM (ours)** | **0.057** | **22.49** | **0.173** | **0.968** | **1.137** | **0.885** |
| A = LightplaneLRM [6] | 0.095 | 18.60 | 0.456 | 0.953 | 1.313 | 0.872 |
| B = A + VolSDF rendering | 0.094 | 20.91 | 0.201 | 0.957 | 1.212 | 0.875 |
| C = B + Direct SDF loss | 0.083 | 21.75 | **0.173** | **0.968** | **1.137** | **0.885** |
| D = C + Texture refinement | **0.057** | **22.49** | **0.173** | **0.968** | **1.137** | **0.885** |

same input views at $512^2$ resolution. Since none of the latter predict PBR materials and since GSO lacks ground-truth PBR materials, for fairness, we use a variant of our model that predicts shaded object textures.

As shown in Figs. 4 and 9 and Tab. 3, our method outperforms all baselines across all metrics. GRM captures texture detail well but struggles with fine geometric structures when meshified. InstantMesh and LightplaneLRM improve geometry but fall short on finer details and texture quality. Our approach excels in reconstructing shapes with detailed geometry and high-fidelity textures.

Ablations in Tab. 3 and Fig. 4 show that incorporating our scalable SDF-based rendering and direct SDF loss into the base LightplaneLRM model enhances geometric quality. Adding texture refinement further brings fine texture details.

Next, we consider the task of **sparse-view reconstruction with PBR materials**, where the goal is to reconstruct the 3D geometry and texture properties (albedo, metalness, and roughness) from four posed shaded 2D views of an object. This is done on an internal dataset of 256 artist-created 3D meshes, curated for high-quality materials. Since there are no existing few-view feed-forward PBR reconstructors, we conduct an ablation study in Tab. 1 and Figs. 3 and 13.

While adding material prediction with additional MLP heads provides some improvements, we observe that incorporating the deferred shading loss and texture refinement is essential for high-quality PBR decomposition. Example PBR predictions are shown in Fig. 8.

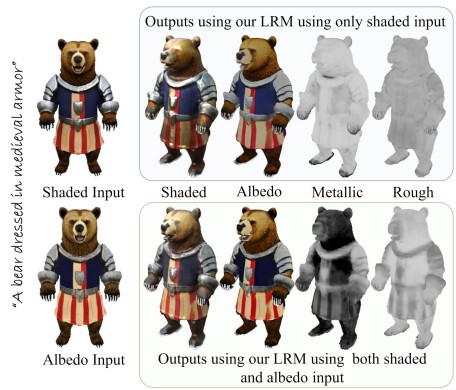

*"A bear dressed in medieval armor"*

Figure 3: **Qualitative ablation on albedo generation.** In text-to-3D, generating 4 views representing albedo colors alongside shaded RGB colors improves material estimation for our 3D reconstructor. With both inputs, the model accurately predicts the armor as metallic and smooth, while the bear's fur is rough.

## 4.2 Text-to-3D generation

Finally, we evaluate text-to-3D with PBR materials. We compare against state-of-the-art feed-forward methods that generate assets at comparable speed ($\approx 10$ to $30$ s per asset). This includes text-to-3D variants of GRM [105], InstantMesh [103], and LightplaneLRM [6]. GRM uses Instant3D's 4-view grid generator, InstantMesh receives the first view from our 2D diffusion model and subsequently generates 6 views, while LightplaneLRM accepts 4 views from our grid generator. Since these methods bake lighting instead of generating PBR materials, for evaluation we apply flat texture

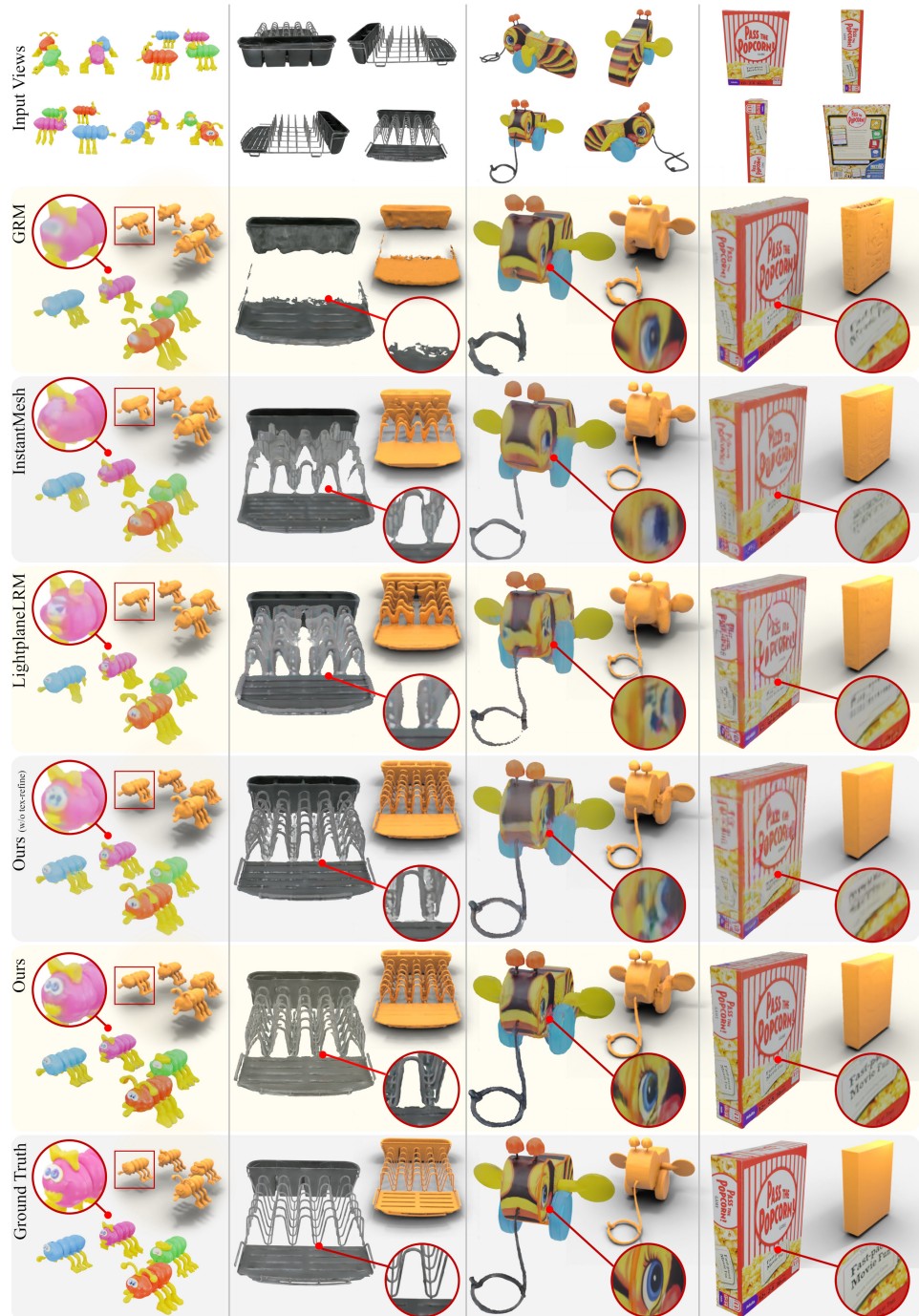

Figure 4: **Qualitative comparison for sparse-view reconstruction.** AssetGen gives better geometry (shown in orange) and higher fidelity texture (inset) compared to state of the art. SDF representation along with the direct SDF loss gives a better geometry compared to the base LightplaneLRM model which uses occupancy (row 4 and 5). Furthermore, our texture refiner greatly enhances texture fidelity (row 5 and 6).

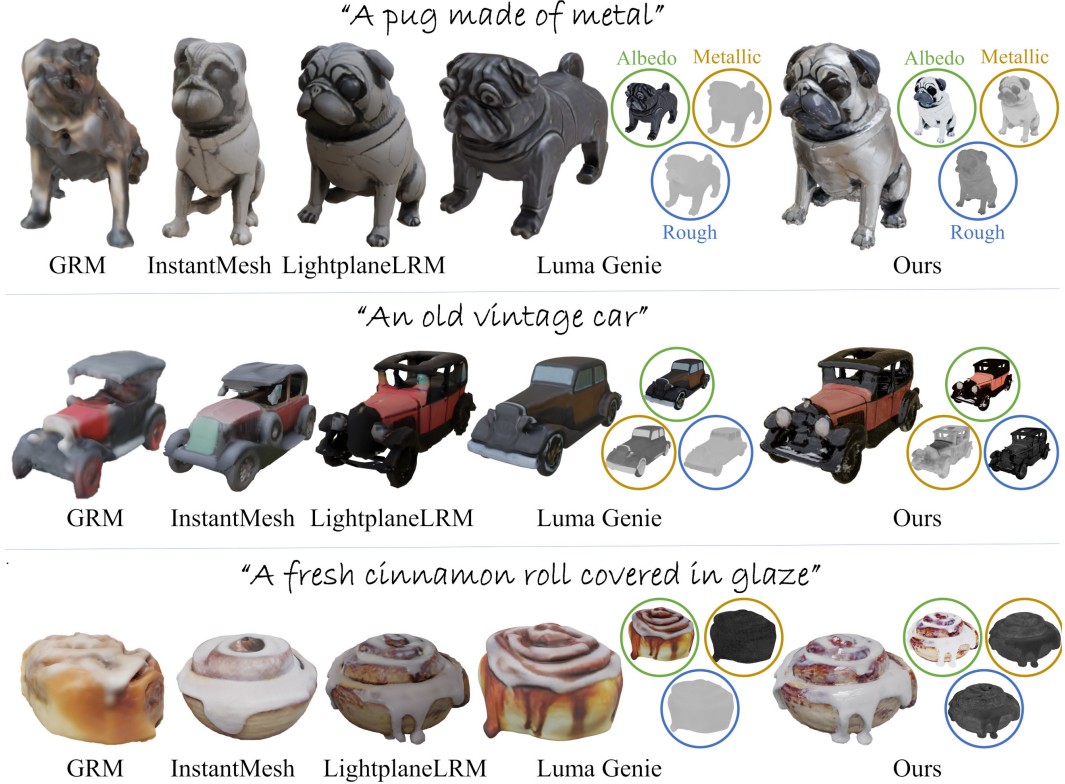

"A pug made of metal"

GRM    InstantMesh    LightplaneLRM    Luma Genie    Ours

"An old vintage car"

GRM    InstantMesh    LightplaneLRM    Luma Genie    Ours

"A fresh cinnamon roll covered in glaze"

GRM    InstantMesh    LightplaneLRM    Luma Genie    Ours

Figure 5: **Qualitative comparison for text-to-3D.** We compare 3D meshes generated by Meta 3D AssetGen and state-of-the-art baselines. We include material decomposition for methods producing PBR materials (Luma Genie and our Meta 3D AssetGen). Our approach produces higher quality materials with better-defined metalness and roughness, and a more accurate decoupling of lighting effects in the albedo.

shading to our outputs. Additionally, we compare with the preview stage of Meshy v3 [85] and LumaAI Genie 1.0 [84], proprietary text-to-3D methods with PBR workflow capable of creating assets within 30 and 15 s respectively. A comparison with the significantly longer refinement stages for Luma and Meshy is provided in the appendix. Fig. 5 shows that AssetGen meshes are visually more appealing and have meaningful materials Figs. 6 and 12 provide more examples and comparisons and showcase fine-grained material control.

For quantitative evaluation, we conducted an extensive user study in Tab. 2 using the 404 deduplicated text prompts from DreamFusion [68]. Users were shown 360° videos of the generated and baseline meshes and were asked to rate them based on 3D shape quality and alignment with the text prompt. A total of 11,080 responses were collected, with significant preference for AssetGen's meshes.

Finally, we ablate the effect of generating dual-channel albedo+shaded grids compared to albedo-only input in Fig. 3 revealing significant PBR decomposition superiority of the former. Additionally, Fig. 13 illustrates the effect of our deferred shading loss.

## 5    Conclusions

We have introduced Meta 3D AssetGen, a significant advancement in sparse-view reconstruction and text-to-3D. Meta 3D AssetGen can generate 3D meshes with high-quality textures and PBR materials faithful to the input text. This uses several key innovations: generating multi-view grids with both shaded and albedo channels, introducing a new reconstruction network that predicts PBR materials from this information, using deferred shading to train this network, improving geometry via a new scalable SDF-based renderer and SDF loss, and introducing a new texture refinement network. Comprehensive evaluations and ablations demonstrate the effectiveness of these design choices and state-of-the-art performance.

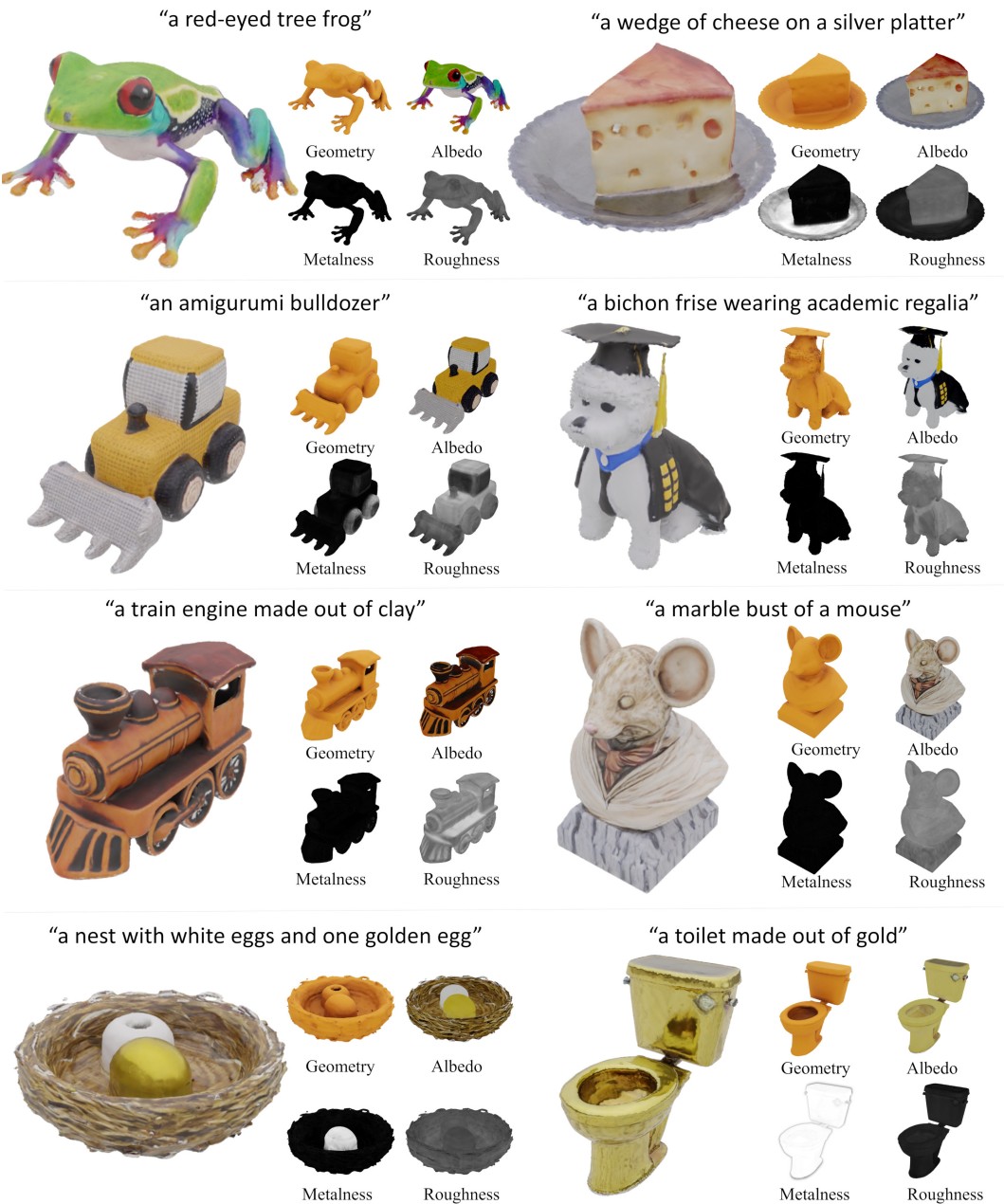

Figure 6: **Text-to-3D** meshes generated by Meta 3D AssetGen along with their PBR decomposition. Note that Meta 3D AssetGen provides detailed albedo and material properties, as highlighted by the metalness of the platter (top right) and the golden objects (last row).

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

# A Appendix

## A.1 Societal Impact

Safeguards should be implemented to prevent abuse, such as filtering input text prompts and detecting unsafe content in generated 3D models. Additionally, our generation process may be vulnerable to biases present in the data and 3D models it relies on, potentially perpetuating these biases in the generated content. Despite these risks, our method can augment the work of artists and creative professionals by serving as a complementary tool to boost productivity. It also holds the potential to democratize 3D content creation, making it accessible to those without specialized knowledge or expensive proprietary software.

## A.2 Limitations

Meta 3D AssetGen significantly advances shape generation but faces several limitations. Despite the fine-tuning of the multiview image grid generator for view consistency, it is not guaranteed, potentially impacting 3D reconstruction quality. Since we use an SDF as an underlying representation, the reconstructor may incorrectly model translucent objects or thin structures like hair or fur. Additionally, while our scalable Triton [67] implementation supports a triplane representation at a resolution of $128 \times 128$, this representation is inefficient, as much of its capacity is used for empty regions. Future work could explore scalable representations such as octrees, sparse voxel grids, and hash-based methods, which may remove the need for a separate texture enhancement model. We also only predict albedo, metalness and roughness, and not emissivity or ambient occlusions. Finally, our method has only been tested on object-level reconstructions, leaving scene-scale 3D generation for future research.

## A.3 Additional qualitative comparisons

This section describes additional qualitative comparisons that, due to limited space, could not be included in the main paper. Firstly, please refer to the video attached in the supplementary material which provides a holistic presentation of Meta 3D AssetGen's qualitative results. In Fig. 7, we highlight the contributions of MetaILRM in geometry, texture and material reconstruction. In Fig. 12, we visualize the control of materials provided by Meta 3D AssetGen, i.e., metalness and roughness, by changing the text prompt for the same concept. Fig. 8 visualizes the renders of the material maps extracted with MetaILRM given four input test views. In Fig. 9, we provide a more extensive qualitative comparison to MeshLRM, the strongest few-view reconstruction baseline. Finally, Fig. 6 provides a gallery of text-conditioned generations depicting Blender-shaded renders together with the rendered PBR maps.

## A.4 User-study details

As described in Sec. 4.2, we conducted an user study on 404 meshes generated using the DreamFusion [68] prompt-set on a standard crowdsourcing marketplace. In the study, users were shown 360°

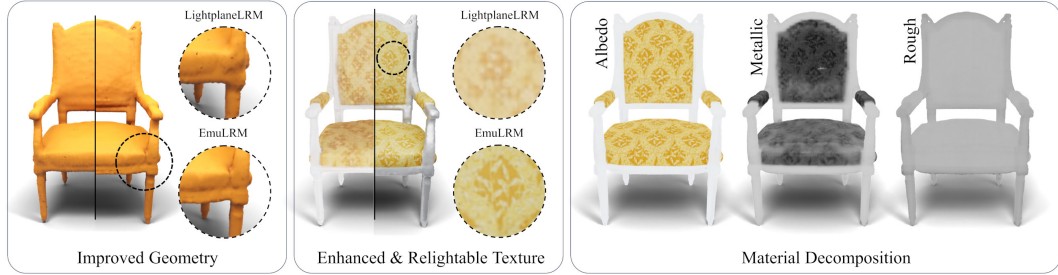

Figure 7: MetaILRM builds upon LightplaneLRM [6], providing improved geometry by employing SDF as a representation, along with direct scalable losses in 3D, improved texture using a UV space texture refiner, and material decomposition by predicting material properties regularized through a novel deferred shading loss.

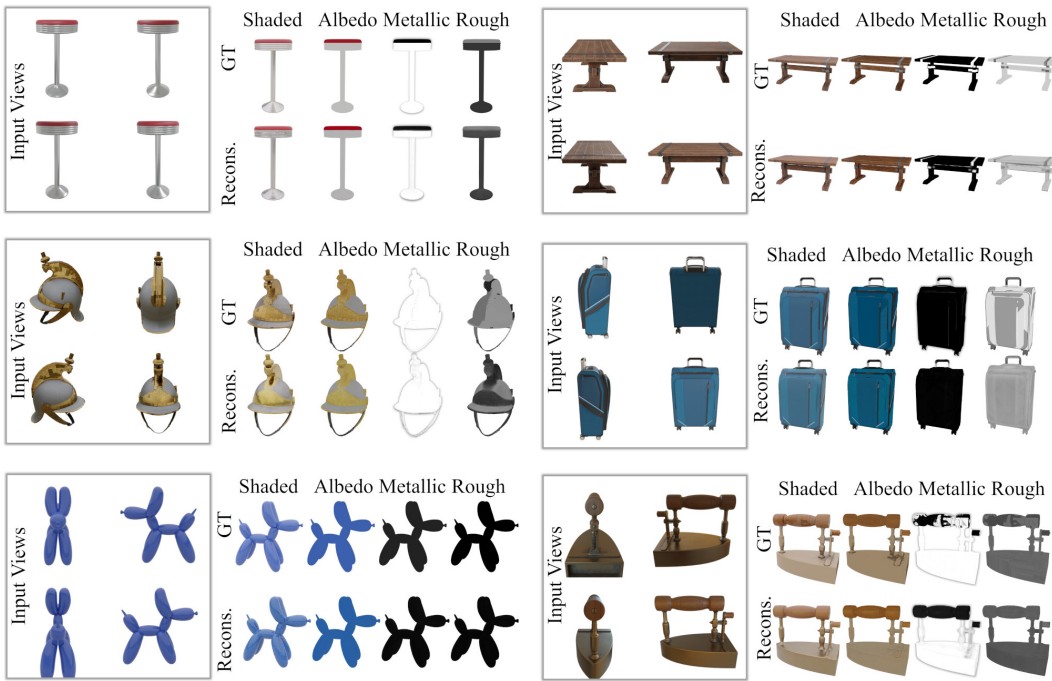

Figure 8: Sparse view reconstruction with intrinsic decomposition. Here the MetaILRM takes 4 shaded views as input and reconstructs the 3D object along with it's albedo, metallic and roughness properties.

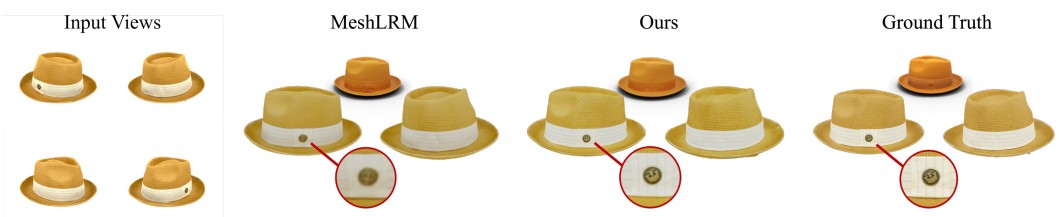

Figure 9: Qualitative comparison on the task of sparse view reconstruction against MeshLRM [97]. Note the higher quality texture detail in our results. Since an open-source implementation of MeshLRM has not yet been released, we compare against the meshes provided on their webpage.

videos of the generated and baseline meshes and were asked to rate them based on 3D shape quality and alignment with the text prompt as shown in Fig. 11. They were asked to consider various factors like identity (whether the object matches what is described in the prompt), texture, existence of Janus problems, and bad geometry (like floaters, disconnected components, etc). 11,080 responses were collected in total, with 5 responses per pair of videos, to eliminate variance in user preference.

### A.5 Additional text-to-3D comparisons

While Tab. 2 compared Meta 3D AssetGen's text-to-3D generations to several fast baselines, for completeness, this section includes additional comparisons to significantly slower methods. More specifically, we conduct the same user study as in Tab. 2 but we compare to the "refinement" stages of the industry baselines Meshy v3 and Luma Genie whose asset generation time is 5 and 10 min re-

Table 4: **Win-rate of Meta 3D AssetGen in text-to-3D user study** evaluating visual quality and text alignment of the generated meshes. In addition to Tab. 2, here we compare to slower baselines. Our Meta 3D AssetGen generates a 3D asset within 30 sec.

| Method | Visual quality | Text fidelity | Runtime |
|---|---|---|---|
| Meshy v3, refined [85] | 77.5 % | 80.9 % | 300 sec |
| Luma Genie 1.0, refined (hi-res) [84] | 51.2 % | 46.3 % | 600 sec |

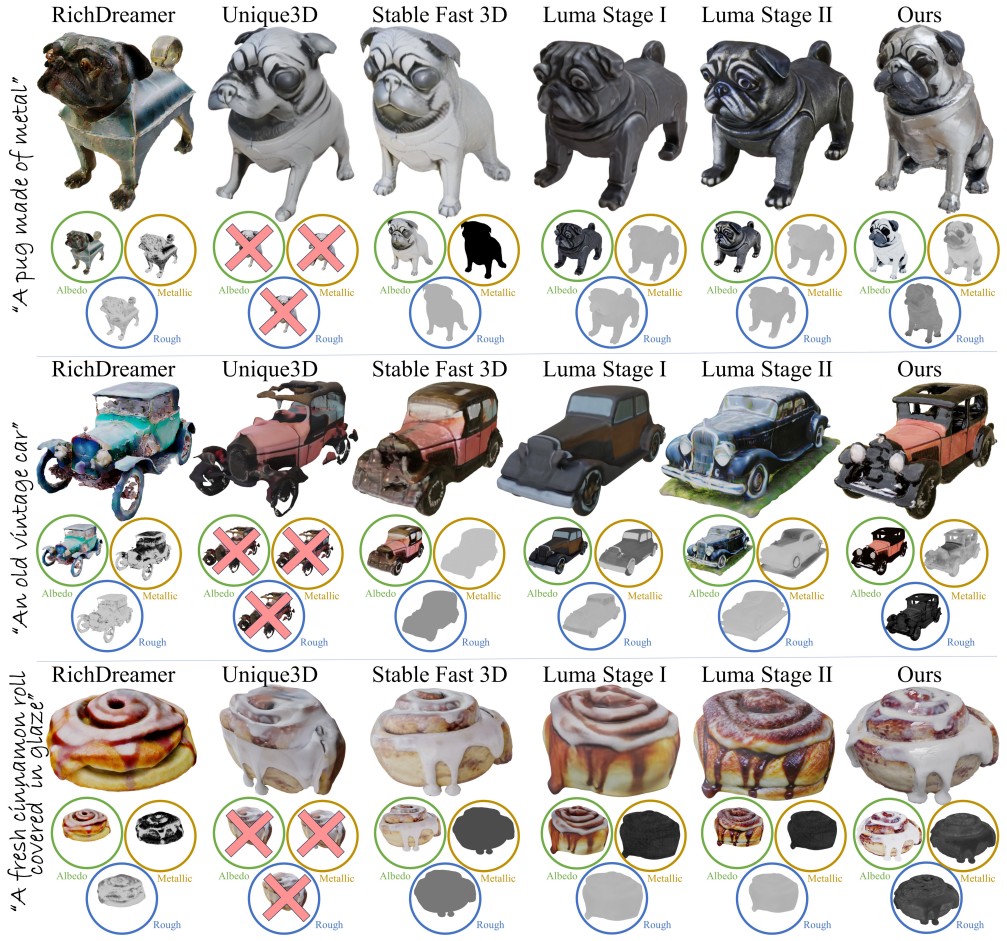

Figure 10: Comparison against RichDreamer, Unique3D, the very recently released Stable Fast 3D Mesh, and Luma Stage 2 (Refinement). RichDreamer takes around an hour per mesh, whereas we generate a mesh in less than 30 seconds with significantly better PBR materials. RichDreamer struggles to separate lighting effects from albedo and generates suboptimal geometry. Unique3D also produces inferior geometry compared to ours and cannot generate PBR materials. Stable Fast 3D Mesh predicts only a single value for metallicity and roughness instead of generating a map. It tends to produce suboptimal geometry and flat objects, as seen with the car and the pug respectively. Luma Stage 2 takes around 10 minutes, generates much better textures than Luma Stage 1, but still struggles to separate illumination from albedo, as evidenced by the car hood.

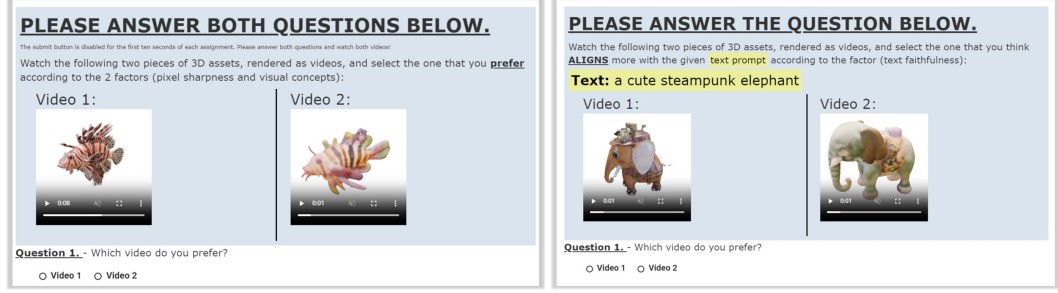

Figure 11: User study interface submitted to a standard crowdsourcing marketplace. Participants are shown videos corresponding to Meta 3D AssetGen and a baseline in a random order, and asked their preference in terms of either quality (left) or faithfulness to the text prompt (right).

spectively. Tab. 4 contains the results of our user-study. Meta 3D AssetGen significantly outperforms Meshy in both text fidelity and visual quality while being $10\times$ faster. Surprisingly, Meta 3D AssetGen is on par with Luma Genie in text fidelity and wins in 40% of cases in visual quality. This is a remarkable result considering Meta 3D AssetGen's $20\times$ better generation time. We show further qualitative comparisons in Fig. 10.

## A.6 Additional technical details

### A.6.1 Grid Generator

We employ a text-to-image diffusion model pre-trained on billions of images annotated with text [16] and expand its input and output channels by a factor of 2 to support simultaneous generation of shaded appearance and albedo. We finetune the model to predict a grid of four images $I_i$, $i = 1, \ldots, 4$, in similar fashion to [74, 42] via minimization of the standard diffusion loss. Training spans a total of 2 days, employing 32 A100 GPUs with a total batch size of 128 and a learning rate of $10^{-5}$.

### A.6.2 MetaILRM

As mentioned in the main paper, MetaILRM is optimized using the direct SDF loss $\mathcal{L}_{sdf}$, PBR loss $\mathcal{L}_{pbr}$, deferred shading loss $\mathcal{L}_{def}$, the binary cross-entropy mask loss $\mathcal{L}_{mask}$, and the depth-MSE loss $\mathcal{L}_{depth}$ so the global objective is:

$$\mathcal{L} = 0.5\mathcal{L}_{sdf} + \mathcal{L}_{pbr} + 0.5\mathcal{L}_{def} + 0.1\mathcal{L}_{mask} + 0.1\mathcal{L}_{depth}.$$

The texture refiner uses only PBR loss and the deferred shading loss: $\mathcal{L}_{pbr} + 0.5\mathcal{L}_{def}$.

In each training batch, we randomly sample 4 views per scene as source input views $I_i$, and another 4 target views $I^{tgt}$, into which we render the sdf-field predicted by MetaILRM, or the mesh predicted by the texture refiner. We then evaluate the aforementioned losses in the target views. 3 scenes per GPU are sampled randomly, and we train on 64 GPUS NVIDIA A100 gpus, yielding an effective batch size of $3 \times 4 \times 64 = 768$ images. The total loss has been optimized using Adam [41] with learning rate $10^{-4}$ for 13K steps.

### A.6.3 Deferred shading loss ablation

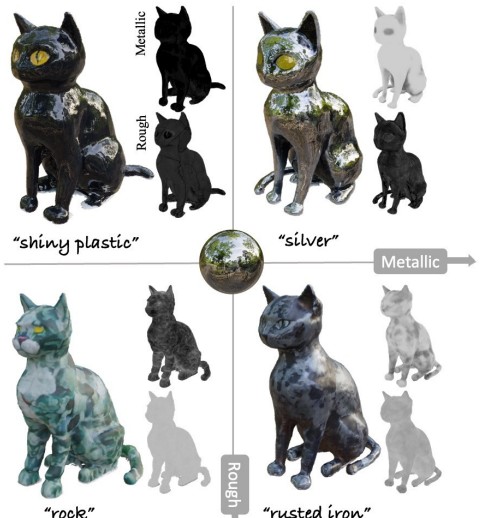

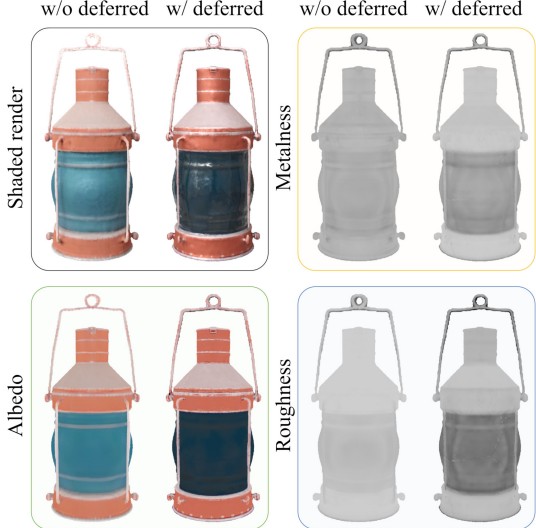

Figure 12: Generated assets for the prompt: "A cat made of <MATERIAL>". Meta 3D Asset-Gen predicts various plausible PBR material maps leading to realistic interaction with the environment light (sphere-mapped in the center)

Figure 13: Using a deferred shading loss on rendered channels enhances PBR quality, resulting in more defined metalness and roughness, such as increased metalness in the lantern's metal parts and decreased roughness in its glass parts.

Besides verifying quantitatively the benefits of the deferred shading loss $\mathcal{L}_{\text{def}}$ in Tab. 1, we also provide a qualitative proof in Fig. 13. Specifically, the PBR materials predicted from albedo&shaded channels exhibit better metalness map on the actual metallic parts of the 3D lantern asset.

### A.6.4 Direct SDF loss $\mathcal{L}_{\text{sdf}}$

We follow Azinovic *et al.*'s [1] direct SDF supervision for the SDF field. Given a pixel $p$ in an image and the sampled points $S_p$ on the ray corresponding to the pixel, the direct SDF loss is computed as

$$\mathcal{L}_{\text{sdf}}(p) = \mathcal{L}_{\text{sdf}}^{\text{tr}}(p) + 0.01\mathcal{L}_{\text{sdf}}^{\text{fs}}(p). \tag{7}$$

$\mathcal{L}_{\text{sdf}}^{\text{fr}}$ is a 'free-space' objective, which forces the MLP to predict a value of 1 for samples $s \in S_p^{\text{fs}}$ which lie between the camera origin and the truncation region of a surface:

$$\mathcal{L}_{\text{sdf}}^{\text{fr}}(p) = \frac{1}{|S_p^{\text{fr}}|} \sum_{s \in S_p^{\text{fs}}} (D_s - 1)^2 \tag{8}$$

where $D_s$ is the predicted SDF from the MLP. For samples within the truncation region ($s \in S_p^{\text{tr}}$), we apply $\mathcal{L}_{\text{sdf}}^{\text{tr}}$, the signed distance objective of samples close to the surface.

$$\mathcal{L}_{\text{sdf}}^{\text{tr}}(p) = \frac{1}{|S_p^{\text{tr}}|} \sum_{s \in S_p^{\text{tr}}} (D_s - \hat{D}_s)^2 \tag{9}$$

A naïve PyTorch implementation of this is memory intensive, because of the evaluation of $B \times H \times W \times N_{\text{ray}}$ points, where $B, H, W, N_{\text{ray}}$ are the number of target images in a batch, the height, width, and the number of points per ray respectively. Therefore, to support large batch sizes, image resolution, and denser point sampling on rays, we implement the direct SDF loss using custom Triton [67] kernels.

### A.6.5 Depth loss $\mathcal{L}_{\text{depth}}$

The depth loss $\mathcal{L}_{\text{depth}}$ minimizes the mean-squared error between the rendered depth prediction $\mathcal{R}_{\text{depth}}(\cdot \mid \hat{s}, \pi)$ the ground-truth depth $\mathcal{R}_{\text{depth}}(\cdot \mid M, \pi)$

$$\mathcal{L}_{\text{depth}} = \left\| \mathcal{R}_{\text{depth}}(\cdot \mid \hat{s}, \pi) - \mathcal{R}_{\text{depth}}(\cdot \mid M, \pi) \right\|^2,$$

where $\mathcal{R}_{\text{depth}}(s, \pi)$ is an operator rendering the depth-map of the shape representation $s$ (mesh or an SDF) from the viewpoint $\pi$.

### A.7 Texture refiner

Having described a high-level overview of our texture refiner in Sec. 3.3, here we provide more details.

As mentioned, the texture refiner network $\Phi$ accepts $N + 1$ texture images $K_i$ in total. The first input to the network is the augmented texture image $K_0 \in \mathbb{R}^{V \times V \times 11}$ given by:

$$\forall v \in [0, V]^2: \quad K_0(v) = \begin{cases} k(\boldsymbol{x}_v) \oplus \boldsymbol{n}(\boldsymbol{x}_v) \oplus \boldsymbol{x}_v, & \text{if } v \in \text{Im}(\phi), \\ \boldsymbol{0}, & \text{otherwise,} \end{cases} \quad \text{where } \boldsymbol{x}_v = \phi(v).$$

The condition $v \in \text{Im}(\phi)$ selects 'valid' UV points that correspond to mesh points; and $\oplus$ denotes channel-wise concatenation, so that $K_0(v)$ is the stack of the 5 PBR parameters $k(\boldsymbol{x}_v)$ from MetaI LRM, normal $\boldsymbol{n}(\boldsymbol{x}_v)$, and the 3D point $\boldsymbol{x}_v$.

In addition to $K_0$, we input to the network $\Phi$ texture images $K_i$, each extracted by looking up information from the corresponding input view $I_i$ directly (thus sidestepping MetaILRM). As noted above, each valid texture point $v$ corresponds to a unique 3D point $\boldsymbol{x}_v = \phi(v) \in M$ on the mesh, which in turn projects to a pixel $u = \pi_i(\boldsymbol{x}_v)$ in the image $I_i$. Let $\chi_i(v) \in \{0, 1\}$ be the flag that tells if point $\boldsymbol{x}_v$ is *visible* in image $I_i$ or not. When point $\boldsymbol{x}_v$ is visible in several views, it is best measured in the most frontal one, which is captured by the cosine $\boldsymbol{\omega}_{\text{o}} \cdot \boldsymbol{n}(\boldsymbol{x}_v)$ between the normal $\boldsymbol{n}$ at $\boldsymbol{x}_v$

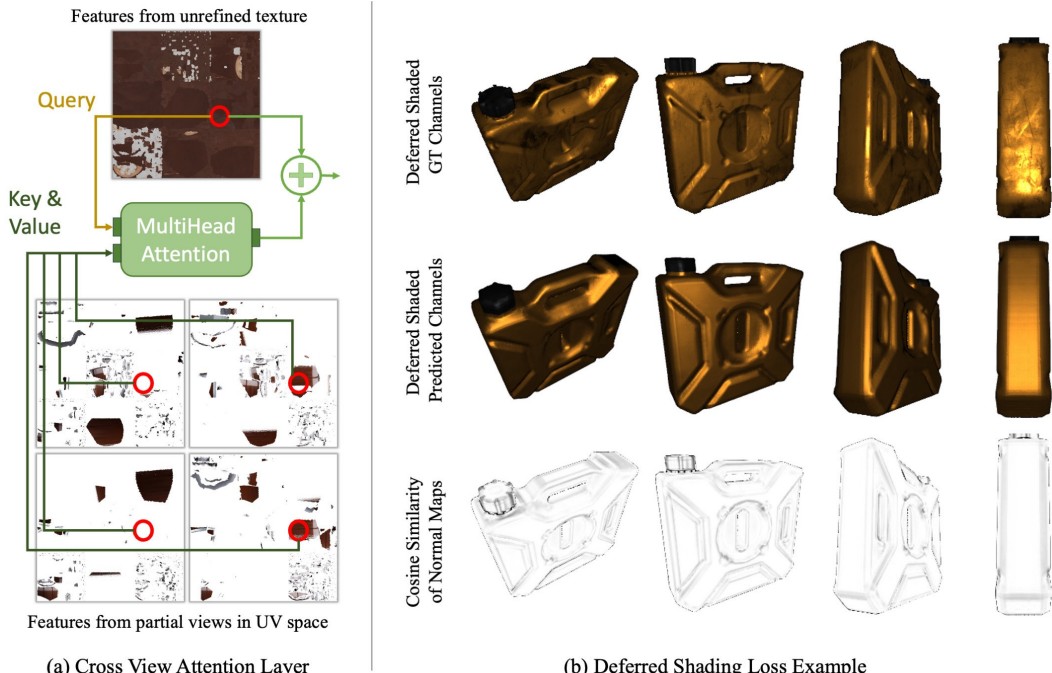

Features from unrefined texture

Query

Key & Value

MultiHead Attention

Features from partial views in UV space

Deferred Shaded GT Channels

Deferred Shaded Predicted Channels

Cosine Similarity of Normal Maps

(a) Cross View Attention Layer

(b) Deferred Shading Loss Example

Figure 14: **(a)** Illustration of Cross-View Attention. Cross-view attention facilitates communication between the UNet branches processing the predicted texture features and the UV space projected input views. This layer blends the predicted texture features with the UV projected input view features based on their match using a multiheaded attention mechanism. **(b)** Example of Deferred Shading Loss Calculation. Deferred shading computes pixel shading using albedo, metalness, roughness, normals, object position, and light source position. We apply it to both the ground truth channels (top) and the predicted channels (middle). The error is calculated as the difference between the two, weighted by the similarity between ground truth normals and predicted normals, to avoid penalizing shading errors due to incorrect normals.

and the ray direction $\boldsymbol{\omega}_v \propto \boldsymbol{x}_0 - \boldsymbol{x}_v$. All this information is packed into additional texture images $K_i \in \mathbb{R}^{V \times V \times (D+1)}$ by setting:

$$\forall v \in [0, V]^2 : \quad K_i(v) = \begin{cases} I_i(\pi_i(\boldsymbol{x}_v)) \oplus (\boldsymbol{\omega}_v \cdot \boldsymbol{n}(\boldsymbol{x}_v)), & \text{if } v \in \text{Im}(\phi) \text{ and } \chi_i(v) = 1, \\ \boldsymbol{0}, & \text{otherwise.} \end{cases}$$

The texture network $\Phi$ is a U-Net that takes as input the texture augmented texture image $K_0$ and outputs the final enhanced texture $K \in \mathbb{R}^{V \times V \times 5}$. This network also fuses information from the view-specific texture images $K_i$. The goal is to select, for each UV point $v$, which of the $N$ input views provides the best information. This is achieved via cross-attention. Specifically, each $K_i$ is processed in parallel by another U-Net, and the first queries information across all the others via multi-head cross attention. In Fig. 14 (a), we provide an illustration of the latter cross-view attention layer.

## A.8 Physically-Based Rendering: Radiance, BRDFs, and models

We briefly summarise key notion of radiometry and standard BRDF models, and then provide a precise expression of the BRDF model used in Meta 3D AssetGen.

### A.8.1 Radiance

The *radiant flux* $\Phi$ is the electromagnetic power flowing through a particular surface $A \subset \mathbb{R}^3$ oriented by the unit normal $\boldsymbol{n}$. The *radiance* $L(p, \boldsymbol{\omega})$ is the radiant flux density at $p$ towards a particular direction $\boldsymbol{\omega}$ per unit *orthogonal* area $dA_\perp$ and per unit solid angle $d\Omega$. The unit vector $\boldsymbol{\omega}$ points at the direction of propagation of the flux.

The flux density is measured with respect to an area which is orthogonal to the direction of propagation $\boldsymbol{\omega}$. In fact, the energy flow is the same through all areas that cut the same 'tube of flux'; the specific area is irrelevant and not a property of the radiation. This dependency is removed by considering the normalized area $dA_\perp$, which is orthogonal to the direction of the flux.

Because the radiance is expressed in units of orthogonal area $dA_\perp$, in order to compute the flux through the surface patch $dA$, which may not be orthogonal, we must account for the *foreshortening factor*, which relates the areas $dA$ and $dA_\perp$:

$$dA_\perp = |\langle \boldsymbol{n}, \boldsymbol{\omega} \rangle|\, dA.$$

With this, the flux that passes through $dA$ towards direction $\boldsymbol{\omega}$ in the solid angle $d\Omega$ is

$$d\Phi = L(p, \boldsymbol{\omega})\,|\langle \boldsymbol{n}, \boldsymbol{\omega} \rangle|\, dA\, d\Omega.$$

Note that $d\Phi$ depends on both $\boldsymbol{\omega}$ and $\boldsymbol{n}$ whereas $L$ only depends on $\boldsymbol{\omega}$. This reinforces the notion that $\boldsymbol{n}$ is a property of the surface, not of the radiation.

## A.9 Reflectance models

Let $p$ be a point on a surface $A$ that separates two media and let $dA$ be a surface patch sitting at $p$. Let $\boldsymbol{n}$ be the normal at this point. We now consider the case where $A$ separates air or empty space from an opaque object, with $\boldsymbol{n}$ pointing towards the outside of this object.

Let $\boldsymbol{\omega}$ be an orientation on the same side of the surface as $\boldsymbol{n}$, i.e., such that $\langle \boldsymbol{n}, \boldsymbol{\omega} \rangle \geq 0$. From the viewpoint of the object, we interpret $L(p, \boldsymbol{\omega})$ as *outgoing radiant flux* and $L(p, -\boldsymbol{\omega})$ as *incoming radiant flux*. These two quantities are related by the *Bidirectional Reflectance Distribution Function* (BRDF) $f$, defined such that:

$$\frac{dL(p, \boldsymbol{\omega}_\mathrm{o})}{d\Omega_\mathrm{i}} = f(p, \boldsymbol{\omega}_\mathrm{i}, \boldsymbol{\omega}_\mathrm{o})\, \langle \boldsymbol{n}, \boldsymbol{\omega}_\mathrm{i} \rangle L(p, -\boldsymbol{\omega}_\mathrm{i}). \tag{10}$$

In this definition, for convenience both $\boldsymbol{\omega}_\mathrm{i}$ and $\boldsymbol{\omega}_\mathrm{o}$ are taken on the same side as $\boldsymbol{n}$ (hence the negative sign in front of $\boldsymbol{\omega}_\mathrm{i}$). A useful consequence is that we do not need to take the absolute value of the inner product $\langle \boldsymbol{n}, \boldsymbol{\omega}_\mathrm{i} \rangle$ as this is positive by definition.

The BRDF thus takes the radiation receives from direction $-\boldsymbol{\omega}_\mathrm{i}$ and distributes it along various outgoing directions $\boldsymbol{\omega}_\mathrm{o}$. Integrating over all incoming directions, gives use the overall radiation reflected towards $\boldsymbol{\omega}_\mathrm{o}$:

$$L(p, \boldsymbol{\omega}_\mathrm{o}) = \int_{H(\boldsymbol{n})} \frac{dL(p, \boldsymbol{\omega}_\mathrm{o})}{d\Omega_\mathrm{i}}\, d\boldsymbol{\omega}_\mathrm{i} = \int_{H(\boldsymbol{n})} f(p, \boldsymbol{\omega}_\mathrm{i}, \boldsymbol{\omega}_\mathrm{o})\, \langle \boldsymbol{n}, \boldsymbol{\omega}_\mathrm{i} \rangle L(p, -\boldsymbol{\omega}_\mathrm{i})\, d\Omega_\mathrm{i}. \tag{11}$$

where $H(\boldsymbol{n}) = \{\boldsymbol{\omega} : \langle \boldsymbol{n}, \boldsymbol{\omega} \rangle \geq 0\}$ is the hemisphere. Next, we provide common basic models for the BRDF function in PBR.

### A.9.1 Diffuse reflectance

In diffuse reflectance, the radiation is absorbed by the material, internally scattered in random directions, and output again to give rise to a uniform distribution. Namely, the *diffuse BRDF* is:

$$f(p, \boldsymbol{\omega}_\mathrm{i}, \boldsymbol{\omega}_\mathrm{o}) = \frac{R}{\pi}$$

where $0 \leq R \leq 1$ is the fraction of power reflected by the diffusion process. The $1/\pi$ factor ensures that the total energy is conserved when $R = 1$.

### A.10 Specular reflectance

The reflection for a perfectly flat interface between two media at $p$ is *specular*: the incoming light radiation $-\boldsymbol{\omega}_\mathrm{i}$ is partially reflected in the specular direction $\boldsymbol{\omega}_\mathrm{o} = r(\boldsymbol{n}, \boldsymbol{\omega}_\mathrm{i}) = 2\boldsymbol{n}\langle \boldsymbol{n}, \boldsymbol{\omega}_\mathrm{i} \rangle - \boldsymbol{\omega}_\mathrm{i}$, and partially transmitted. This phenomena is characterised by *Fresnel's equations*, which are derived from Maxwell's equations, utilising continuity conditions for the electromagnetic field at the interface between the two media. Fresnel's equations describe the planar radiation in full, including its polarisation (in the most general case, using phasors, and thus complex numbers). For graphics, we

assume that light is unpolarized, so we only calculate the power. The fraction of power reflected is given by Fresnel's coefficient (using Schlick's approximation [72]):

$$F(\langle \boldsymbol{n}, \boldsymbol{\omega}_i \rangle) = F_0 + (1 - F_0) |\langle \boldsymbol{n}, \boldsymbol{\omega}_i \rangle|^5, \quad F_0 = \left( \frac{\hat{n}_1 - \hat{n}_2}{\hat{n}_1 + \hat{n}_2} \right)^2.$$

Here $\hat{n}_1$ and $\hat{n}_2$ are the indices of reflectivity of the two media, respectively. This equation is valid for dielectrics (non-metallic objects), but also used as an approximation for metals by tweaking $F_0$.

In order to write this relation as a BRDF, we write:

$$L(p, \boldsymbol{\omega}_o) = \int_{H(\boldsymbol{n})} f(p, \boldsymbol{\omega}_i, \boldsymbol{\omega}_o) \langle \boldsymbol{n}, \boldsymbol{\omega}_i \rangle L(p, -\boldsymbol{\omega}_i) \, d\Omega_i = R(\langle \boldsymbol{n}, r(\boldsymbol{n}, \boldsymbol{\omega}_o) \rangle) L(p, -r(\boldsymbol{n}, \boldsymbol{\omega}_o)).$$

Hence, the BRDF must be a delta function centered at $\boldsymbol{\omega}_i^* = r(\boldsymbol{n}, \boldsymbol{\omega}_o)$:

$$f(p, \boldsymbol{\omega}_i, \boldsymbol{\omega}_o) = \frac{F(\langle \boldsymbol{n}, \boldsymbol{\omega}_o \rangle)}{\langle \boldsymbol{n}, \boldsymbol{\omega}_o \rangle} \delta_{r(\boldsymbol{n}, \boldsymbol{\omega}_o)}(\boldsymbol{\omega}_i) \tag{12}$$

where we used the fact that $\langle \boldsymbol{n}, r(\boldsymbol{n}, \boldsymbol{\omega}_o) \rangle = \langle \boldsymbol{n}, \boldsymbol{\omega}_o \rangle$ and where $\delta_{r(\boldsymbol{n}, \boldsymbol{\omega}_o)}$ is the delta distribution centered at $r(\boldsymbol{n}, \boldsymbol{\omega}_o)$.

## A.11 Microfacet models

Rough surfaces can be though of as a collection of randomly-oriented flat microfacets, each reflecting light in a specular fashion. Consider a point $p$ on a surface and incoming and outgoing radiation directions $\boldsymbol{\omega}_i$ and $\boldsymbol{\omega}_o$. If the point contains a microfacet that enables light to reflect from $\boldsymbol{\omega}_i$ to $\boldsymbol{\omega}_o$, then the normal of the microfacet must be $\boldsymbol{m} \propto \boldsymbol{\omega}_i + \boldsymbol{\omega}_o$. If the microfacet is oriented elsewhere, then no light flows in the direction $\boldsymbol{\omega}_o$. Hence, we can write

$$\boldsymbol{m} = h(\boldsymbol{\omega}_i, \boldsymbol{\omega}_o) = \frac{\boldsymbol{\omega}_i + \boldsymbol{\omega}_o}{|\boldsymbol{\omega}_i + \boldsymbol{\omega}_o|}$$

as a function of the incoming and outgoing radiation. This is called *half vector* as it sits in between the two vectors $\boldsymbol{\omega}_i$ and $\boldsymbol{\omega}_o$.

Now we wish to derive the macro BRDF $f(p, \boldsymbol{\omega}_i, \boldsymbol{\omega}_o)$ from the micro BRDF $F(p, \boldsymbol{\omega}_i, \boldsymbol{\omega}_o | \boldsymbol{m})$, where we have emphasized the fact that the BRDF is oriented relative to the microfacet normal $\boldsymbol{m}$. A short calculation [90] shows that:

$$f(p, \boldsymbol{\omega}_i, \boldsymbol{\omega}_o) = f_m(p, \boldsymbol{\omega}_i, \boldsymbol{\omega}_o | \boldsymbol{m}) \frac{\langle \boldsymbol{m}, \boldsymbol{\omega}_i \rangle}{\langle \boldsymbol{n}, \boldsymbol{\omega}_i \rangle} \frac{\langle \boldsymbol{m}, \boldsymbol{\omega}_o \rangle}{\langle \boldsymbol{n}, \boldsymbol{\omega}_o \rangle} \frac{1}{\langle \boldsymbol{m}, \boldsymbol{n} \rangle}.$$

In practice, there is a distribution over possible surface normals $\boldsymbol{m}$, characterised by the *microfacet distribution function* $D(\boldsymbol{m} | \boldsymbol{n})$. The latter is defined such that $D(\boldsymbol{m} | \boldsymbol{n}) \, dA \, d\Omega_m$ is the total area of the microfacets within patch $dA$ of the macrosurface with orientation in $d\Omega_m$. In practice, only part of the microfacet is visible and illuminated, depending on the interaction with other facets. This is accounted for by the *shadowing-masking* function $G(\boldsymbol{\omega}_i, \boldsymbol{\omega}_o, \boldsymbol{m}, \boldsymbol{n}) \in [0, 1]$. The expected reflectance is thus:

$$f(p, \boldsymbol{\omega}_i, \boldsymbol{\omega}_o) = \int_{H(\boldsymbol{n})} \frac{\langle \boldsymbol{m}, \boldsymbol{\omega}_i \rangle}{\langle \boldsymbol{n}, \boldsymbol{\omega}_i \rangle} \frac{\langle \boldsymbol{m}, \boldsymbol{\omega}_o \rangle}{\langle \boldsymbol{n}, \boldsymbol{\omega}_o \rangle} f_m(p, \boldsymbol{\omega}_i, \boldsymbol{\omega}_o | \boldsymbol{m}) \, D(\boldsymbol{m} | \boldsymbol{n}) \, G(\boldsymbol{\omega}_i, \boldsymbol{\omega}_o, \boldsymbol{m}, \boldsymbol{n}) \, d\Omega_m.$$

Plugging Eq. (12) in the value for the mirror-like reflectance $f_m$ for each microfacet, we get [90]:

$$f(p, \boldsymbol{\omega}_i, \boldsymbol{\omega}_o) = \frac{F(\langle \boldsymbol{h}, \boldsymbol{\omega}_o \rangle) D(\boldsymbol{h} | \boldsymbol{n}) \, G(\boldsymbol{\omega}_i, \boldsymbol{\omega}_o, \boldsymbol{h}, \boldsymbol{n})}{4 \langle \boldsymbol{n}, \boldsymbol{\omega}_i \rangle \langle \boldsymbol{n}, \boldsymbol{\omega}_o \rangle} \quad \text{where } \boldsymbol{h} = h(\boldsymbol{\omega}_i, \boldsymbol{\omega}_o).$$

## A.12 Standard microfacet models

Here, we discuss common choices for the functions $D$ and $G$ in standard PBR models.

**The Torrance-Sparrow model.** The oldest such model is due to Torrance and Sparrow [87]. They simply assume a Gaussian model for the microfacet distribution function:

$$D(\boldsymbol{m}|\boldsymbol{n}) = b\exp\left(-\alpha^2\theta\right), \quad \text{where } \cos\theta = \langle\boldsymbol{m},\boldsymbol{n}\rangle,$$

and $b$ is a suitable normalization constant. For the shadowing-masking function they pick:

$$G(\boldsymbol{\omega}_\mathrm{i}, \boldsymbol{\omega}_\mathrm{o}, \boldsymbol{m}, \boldsymbol{n}) = \min\left\{1, \frac{2\langle\boldsymbol{m},\boldsymbol{n}\rangle\langle\boldsymbol{n},\boldsymbol{\omega}_\mathrm{i}\rangle}{\langle\boldsymbol{m},\boldsymbol{\omega}_\mathrm{i}\rangle}, \frac{2\langle\boldsymbol{m},\boldsymbol{n}\rangle\langle\boldsymbol{n},\boldsymbol{\omega}_\mathrm{o}\rangle}{\langle\boldsymbol{m},\boldsymbol{\omega}_\mathrm{o}\rangle}\right\}.$$

This function has a simple geometric derivation under a basic geometric model of the microfacets.

**The Beckmann-Spizzichino-Smith model.** Beckmann and Spizzichino [2] suggested the model:

$$D(\boldsymbol{m}|\boldsymbol{n}) = \frac{1}{\pi\alpha^2\cos^4\theta}e^{-\frac{\tan^2\theta}{\alpha^2}} \quad \text{where } \cos\theta = \langle\boldsymbol{m},\boldsymbol{n}\rangle.$$

Smith [77] noted that the shadowing-masking function should be derived from the microfacet distribution function, which describes the micro-geometry of the surface. They also proposed a factorized model $G(\boldsymbol{\omega}_\mathrm{i}, \boldsymbol{\omega}_\mathrm{o}, \boldsymbol{m}, \boldsymbol{n}) = G_1(\boldsymbol{\omega}_\mathrm{i}, \boldsymbol{n})G_1(\boldsymbol{\omega}_\mathrm{o}, \boldsymbol{n})$. For Beckmann's distribution, the Smith shadowing-masking function is given by:

$$G_1(\boldsymbol{\omega}, \boldsymbol{n}) = \frac{2}{1 + \mathrm{erf}(a) + \frac{1}{a\sqrt{\pi}}e^{-a^2}} \quad \text{where } a = \frac{1}{\alpha\tan\theta_{\boldsymbol{\omega}}} \text{ and } \cos\theta_{\boldsymbol{\omega}} = \langle\boldsymbol{n},\boldsymbol{\omega}\rangle.$$

**The GGX model.** The GGX model by [90] is a variant of the Beckmann model, with slightly different microfacet distribution and shadowing-masking function:

$$D(\boldsymbol{m}|\boldsymbol{n}) = \frac{\alpha^2}{\pi\cos^4\theta(\alpha^2 + \tan^2\theta)^2}, \qquad G_1(\boldsymbol{\omega}, \boldsymbol{n}) = \frac{2}{1 + \sqrt{1 + \alpha^2\tan^2\theta_{\boldsymbol{\omega}}}}, \qquad (13)$$

where $\cos\theta = \langle\boldsymbol{m},\boldsymbol{n}\rangle$ and $\cos\theta_{\boldsymbol{\omega}} = \langle\boldsymbol{n},\boldsymbol{\omega}\rangle$.

### A.12.1 The BRDF model used in Meta 3D AssetGen

The BRDF model used in our paper combines diffuse and GGX BRDFs:

$$f(\boldsymbol{\omega}_\mathrm{i}, \boldsymbol{\omega}_\mathrm{o}|k(\boldsymbol{x}), \boldsymbol{n}) = \frac{R}{\pi} + \frac{F(\boldsymbol{h}|\boldsymbol{n})D(\boldsymbol{h}|\boldsymbol{n})G_1(\boldsymbol{n}, \boldsymbol{\omega}_\mathrm{i})G_1(\boldsymbol{n}, \boldsymbol{\omega}_\mathrm{o})}{4(\boldsymbol{n}\cdot\boldsymbol{\omega}_\mathrm{i})(\boldsymbol{n}\cdot\boldsymbol{\omega}_\mathrm{o})}.$$

The first term is the *diffuse component* (Lambertian reflection), where $R \in [0, 1]$ is the fraction of light power reflected by diffusion. The second term in the *specular component*, where $F$ is Schlick's approximation [72] $F(\boldsymbol{\omega}_\mathrm{i}|\boldsymbol{n}) = F_0 + (1 - F_0)(1 - \boldsymbol{n}\cdot\boldsymbol{\omega}_\mathrm{i})^5$ of Fresnel's reflectance and where $F_0 \in [0, 1]$ is the Fresnel coefficient at normal incidence. The unit vector $\boldsymbol{h} \propto \boldsymbol{\omega}_\mathrm{i} + \boldsymbol{\omega}_\mathrm{o}$ is the *half vector*, which is the orientation needed to reflect $\boldsymbol{\omega}_\mathrm{i}$ into $\boldsymbol{\omega}_\mathrm{o}$ by the rough material's microfacets (generally different from but averaging to $\boldsymbol{n}$). The function $D$ and $G_1$ are the microfacet distribution function and the shadowing-masking function given by:

$$D(\boldsymbol{m}|\boldsymbol{n}) = \frac{\alpha^2}{\pi((\boldsymbol{m}\cdot\boldsymbol{n})^2(\alpha^2 - 1) + 1)^2}, \quad G_1(\boldsymbol{n}, \boldsymbol{\omega}) = \frac{2(\boldsymbol{n}\cdot\boldsymbol{\omega})}{\boldsymbol{n}\cdot\boldsymbol{\omega} + \sqrt{(\boldsymbol{n}\cdot\boldsymbol{\omega})^2(\alpha^2 - 1) + \alpha^2}}.$$

These are the same as Eq. (13) with the trigonometric functions expanded in terms of dot products. In this model, the reflectance $R$ and reflectance at normal incidence $F_0$ are RGB triplets. The specular highlight color $F_0$ is approximately white (equal to $\mathbf{1}$) for dielectrics, and colored for metals; furthermore, metals have no diffuse component ($R = \mathbf{0}$). Thus, we introduce the parameter *metalness* $\gamma \in [0, 1]$ and the *base color* $\rho_0 \in [0, 1]^3$ and define:

$$R = \rho_0(1 - \gamma), \qquad F_0 = \mathbf{1}(1 - \gamma) + \rho_0\gamma.$$

In this manner, the albedo is used as diffuse color or specular color depending on whether the material is a dielectric or a metal. In the paper, we call the parameter $\rho_0$ *albedo* as it is a better known concept; however, in this model $\rho_0$ is the albedo only when $\gamma = 0$, i.e., when the object is a perfect dielectric.

The BRDF is thus fully described by the albedo $\rho_0$, the roughness $\alpha$, and the metalness $\gamma$, for a total of five scalar parameters. Hence, the LRM predicts the triplet $k(\boldsymbol{x}) = (\rho_0, \gamma, \alpha)$ at each 3D point $\boldsymbol{x}$.

