# OpenReview forum: "Meta 3D AssetGen: Text-to-Mesh Generation with High-Quality Geometry, Texture, and PBR Materials"
_NeurIPS.cc/2024/Conference — NeurIPS 2024 poster_

### Official Review · Reviewer_Xevg · 2024-07-12

**Soundness:** 3
**Presentation:** 4
**Contribution:** 3
**Rating:** 6
**Confidence:** 4

**Summary:**

Emu3D is a novel approach in text-conditioned 3D generation, producing high-quality 3D meshes and materials using Physically-Based Rendering (PBR). It uses a two-stage process: first generating images from standard viewpoints, then reconstructing the 3D shape and appearance. This approach is faster and more reliable than earlier methods and creates realistic 3D graphics by modeling light interaction with surfaces. Emu3D also improves mesh generation by predicting a signed-distance field (SDF) for better quality and refining textures for greater detail. It surpasses existing methods in both image-to-3D and text-to-3D tasks, offering superior visual quality and text alignment.

**Strengths:**

The quality of the generated geometry, texture as well as the PBR materials offers an efficient and high quality generation workflow. The paper is well-presented, with sufficient experiments to validate the quality of the results.

**Weaknesses:**

- The generated PBR material includes only limited components: albedo, metallicity, and roughness. PBR materials typically also consist of other components, such as index of reflection, scattering, coat, sheen etc. Therefore, the generated material can only represent surface reflectance from diffuse to specular.

**Questions:**

- What’s the performance of the proposed method over anisotropy materials?
- What's the performance of the extracted mesh over transparent object like a class bottle?

**Limitations:**

- The generated material has significant representational limitations, as it primarily models surface reflection using only albedo, metalness, and roughness.

---

> ### Author Rebuttal · Authors · 2024-08-06
>
> We thank the reviewer for the review and the feeback!
>
> 1. **Lack of support for material components such as index of reflection, scattering, coat, sheen etc.**:
>
> Our model indeed only tackles essential PBR parameters (albedo, roughness and metallness) as these are the most important ones, and also the focus of 3D artists when they work on real-time applications like video games. These real-time applications are also the most likely to benefit from AI-generated content as they require a large quantity of 3D assets and, depending on the game, the quality of these assets need not to be as high as, say, a CG for a Hollywood production. Additionally, simple PBR textured assets can serve as a starting point for artists to extend the material map, making it a good place to start from.
>
> Note that in the field of generative 3D asset creation, to our knowledge, there are no methods that go beyond the simple PBR model (albedo, metallness, roughness). Our approach can in principle be extended to predict additional parameters (e.g., say the full Disney BRDF), given sufficiently representative training data.
>
> We’ll add a note in the final version to clarify these limitations.
>
>
> 2. **Performance on anisotropic and glass like materials.**
>
> We also do not tackle transparent materials yet. Our approach could also be extended to those in principle, again given sufficient training data, but less trivially as it would require modifying the deferred shading part to support transparency.

---

> > ### Author Response · Authors · 2024-08-12
> >
> > With the reviewer-author discussion period nearing its end, we want to ensure that our responses have thoroughly addressed your primary concerns. We recognize the importance of this dialogue in resolving the issues raised by the reviewers and are committed to providing any further clarifications you might need. Please let us know if there are any additional questions or if more explanation is required.
> >
> > Warm regards,
> > Emu3D Authors

---

### Official Review · Reviewer_cPQZ · 2024-07-12

**Soundness:** 4
**Presentation:** 4
**Contribution:** 3
**Rating:** 6
**Confidence:** 4

**Summary:**

The paper adopts a two-stage PBR information generation method. In the first stage, the text-to-image model is used to predict the PBR channel. In the second stage, it uses SDF to reconstruct the geometry and PBR information. Then, a cross-view attention texture reflector network is used to improve the coarse UV of the input.

**Strengths:**

Methods used are quite effective. The writing is relatively easy to understand.
The experiments are comprehensive. The performance is indeed quite good.

**Weaknesses:**

The technical novelty is weak.

I noticed that neither the dataset nor the code are intended to be open sourced. I understand that this technology involves commercial use, but I believe that at least all the test sets used in the experiments and the 3D (PBR) assets generated on the test sets should be open sourced to facilitate comparison in future work.

The PBR resolution of 512x512 with 2x2 grid for 4-view is still relative low.

**Questions:**

Recently, Unique3d [1] generated mesh with higher-resolution texture. As Emu3d and Unique3d [1] are concurrent works, it is not necessary to directly compare with it. However, theoretically speaking, compared to Unique3d, the texture resolution of Emu3d is still relatively low. It is suggested to add a paragraph in the paper to discuss the advantages and disadvantages of the two methods relative to each other.

[1] Wu, Kailu, et al. "Unique3D: High-Quality and Efficient 3D Mesh Generation from a Single Image." arXiv preprint arXiv:2405.20343 (2024).

**Limitations:**

No. But they have discussed in the limitation part. They use a triplane representation, which cannot represent large-scale scenes.

---

> ### Author Rebuttal · Authors · 2024-08-06
>
> Thank you for the review and the helpful comments!
>
> 1. **Technical Novelty**:
>
> To our knowledge, Emu3D is one of the first academic work that performs text-to-3D with PBR in a feed-forward way. We introduce several novel aspects that come together to achieve high quality and efficient text-to-3D reconstruction with PBR materials. These include:
>
> (i) A novel albedo + shaded RGB grid generation paradigm, with the intuition that the combination of these two modalities can help the reconstruction stage in inferring materials (ablated in Table 1, Fig. 3 and Video).
>
> (ii) A novel deferred shading loss that regularizes the predicted materials in an efficient manner (ablated in Table 1, A.6.3, Fig. 12 and Video).
>
> (iii) A Volumetric SDF renderer that extends Lightplane with Triton kernels for supporting large batch sizes, image resolution, and denser point sampling on rays. Additionally, we enable a direct loss on the SDF values (A.6.4), also implemented as Triton kernels for scalability (ablated in Table 3, Fig. 4 Row 4 vs 5 and Video).
>
> (iv) A novel texture refinement module that uses cross-view attention layers, which greatly improves the fidelity of reconstructed textures by operating in UV space (ablated in A.7 and Fig. 14, Fig. 4 Row 5 vs Row 6, Table 1 and Table 3, and Video).
>
>
> 2. **Facilitating Comparisons**:
>
> This is a great point and we are in discussion to open the data as much as possible. We also plan to release the PBR assets generated on the dreamfusion prompts, which are a typical benchmark for comparing text to 3D models.
> Note, however, that we do use publicly available GSO dataset for sparse-view image to 3D reconstruction. The reconstructed assets using our method on GSO will also be released.
>
>
> 3. **PBR resolution**:
>
> The resolution of our 2x2 grid is *not* 512x512, it is 512x512 per image in the grid, making the grid resolution to be 1024x1024 (albedo and shaded, each 1024 x 1024 x 3). These images are input to EmuLRM, which produces UV textures with resolution 512x512 each for albedo, metalness and roughness. These are further upscaled to 1024x1024 resolution textures by the texture refiner.
>
>
> 4. **Comparison to Unique3D**:
>
> Unique3D upscales the 4 view grid to a resolution of 2048x2048, compared to our 512x512 per image resolution, which can potentially impart higher detailed textures. However, Unique3D only predicts RGB colors, not texture maps, and lacks material outputs as well. This prohibits asset use in novel environments due to baked in lighting in the albedo.
>
> Further, we show a qualitative comparison to more recent methods like Unique3D as well as Stable 3D which came out last week in the common response PDF. The quality of assets produced by our method is visually more appealing than both.
>
> We will add this discussion and the comparisons to the final draft.

---

> > ### Author Response · Authors · 2024-08-12
> >
> > With the reviewer-author discussion period nearing its end, we want to ensure that our responses have thoroughly addressed your primary concerns. We recognize the importance of this dialogue in resolving the issues raised by the reviewers and are committed to providing any further clarifications you might need. Please let us know if there are any additional questions or if more explanation is required.
> >
> > Warm regards,
> > Emu3D Authors

---

> > ### Comment · Reviewer_Zgd2 · 2024-08-13
> >
> > Thank the authors for the detailed rebuttal and the additional results provided. It’s surprising to see that the texture refinement is effective, even with UV texture maps that contain numerous isolated pieces.
> >
> > Regarding the third point, I meant to ask about optimizing with respect to the renderings of volSDF, rather than the input images. Nevertheless, I agree with the authors’ point regarding the time cost.
> >
> > The rebuttal has addressed my major concerns, and I have adjusted my scores accordingly.

---

### Official Review · Reviewer_Zgd2 · 2024-07-13

**Soundness:** 3
**Presentation:** 4
**Contribution:** 3
**Rating:** 6
**Confidence:** 4

**Summary:**

This paper introduces a two-stage 3D asset generation pipeline that outputs high-quality 3D meshes with PBR materials in approximately 30 seconds. The technical novelties include:
- In text-to-image stage, generating multiple views of both shaded and unshaded images, and predicting PBR materials during the image-to-3D stage.
- Using SDF instead of opacity field which yields higher-quality meshes.
- Introducing a new texture refiner designed to recover and enhance details in texture maps.

**Strengths:**

The proposed pipeline introduces several components to enhance and extend the existing pipeline [39]. Most design choices are intuitive and well-justified through ablation studies, which include both qualitative and quantitative analyses.

Additionally, the paper compares the proposed pipeline with SOTA methods, demonstrating more favorable results through a user study.

**Weaknesses:**

I found certain technical novelties challenging to assess:
- The use of SDF instead of opacity fields in the reconstruction pipeline is not a novel idea but more of an extension to Instant3D. This approach has also been explored in 3D generation works such as MeshLRM and Latte3D.
- The texture refiner, which I found the most interesting and novel part of the pipeline, is difficult to evaluate from the provided qualitative examples. Looking at the texture maps in Figures 2 and 13, which contain many small components and seams from xAtlas outputs, one concern is whether these artifacts might affect the UNet's predictions in UV space, e.g. causing color inconsistencies. The ablation in Figure 14 only shows textures improved in flat regions, without the corresponding UV maps. Showing texture maps before and after refinement around seams would help readers better understand the model's effectiveness, and justify refinement in UV space as opposed to image space.

**Questions:**

Regarding the blurriness of the pre-refinement texture map: Is this a result of the image-to-3D reconstruction process or an issue with sampling from the texture field? If it is due to the sampling process, why not optimizing with respect to volSDF renderings to recover the details?

**Limitations:**

The authors discuss the limitations of their method and broader impact in the paper.

---

> ### Author Rebuttal · Authors · 2024-08-06
>
> We thank the reviewer for the review and the feedback on our work!
>
> 1. **Use of SDF instead of opacity fields:**
>
> We do not claim that using SDF in 3D reconstruction or generation is novel per se (e.g., StyleSDF, SDFusion, GET3D, Latte3D, etc. uses it).
>
> We do note that we are among the first to use SDF in LRMs specifically. Other works that do the same include MeshLRM, InstantMesh, Large Tensorial Model (LDM) and Direct3D, but these are concurrent (released either after or just before the submission deadline).
>
> What makes our approach stand out, even among these more recent papers, is the fact that we use the SDF formulation directly, in a single “phase”. Other papers usually require a first stage training with opacity fields, followed by a second stage where the opacity field is replaced with SDFs / meshes. In other words, we show that two stages are unnecessary.
>
> Furthermore, we do not simply exchange NeRF opacity fields with a VolSDF formulation. Instead, we implement the SDF renderer by extending the open source Lightplane Triton kernels supporting large batch sizes, image resolution, and denser point sampling on rays. Additionally, we enable a direct loss on the SDF values (A.6.4), also implemented as Triton kernels for scalability (note that a naive pytorch implementation of such a loss only allows a batch size of 1 image on an a100 GPU at our base resolution due to increase memory requirements). While these are “engineering” contributions, they are key to making this a practical method.
>
> Note that we compare against the concurrent works MeshLRM and InstantMesh, which were released before our submission.
>
>
> 2. **Texture refinement evaluation, before and after refinement outputs:**
>
> Our texture refiner is evaluated extensively both quantitatively and qualitatively:
>
> (i) *Table 1*: Qualitative evaluation of 4-view PBR reconstruction on the internal dataset. The effectiveness of the texture refiner is evident from improvements in LPIPS and PBR PSNR from config F to config G, where the only change is the addition of the texture refiner. Further improvement from config H to I, again solely due to the texture refiner, underscores its efficacy.
>
> (ii) *Table 3*: Qualitative evaluation of 4-view non-PBR reconstruction on the GSO dataset. The significant improvement in texture quality, particularly in PSNR and LPIPS, between config C and D, demonstrates the impact of adding the texture refiner.
>
> (iii) *Fig. 4, row 5 vs. row 6 (ours without tex-refiner and ours with tex-refiner)*: Qualitative evaluation shows the effectiveness of the texture refiner, with clear improvements in texture fidelity.
>
> (iv) *Fig. 14*: Qualitative evaluation further demonstrating the impact of texture refinement.
>
> (v) *Supplementary Video*: 360 degree view texture refinement ablation for select assets.
>
> For before and after refinement visualizations of uv texture maps, please check the common PDF doc. We will further add UV space improvements to the final draft for better understanding as suggested.
>
>
> 3. **Blurriness in Image-to-3D and why not optimization:**
>
> We suspect that the degradation in textures is due to the volumetric 3D representation of colors. Since texture is inherently a surface property, extending it into 3D space beyond the surface is inefficient, leading to blurriness.
>
> For instance, this degradation occurs in all volumetric LRM-based methods like Instant3D, Instant Mesh, TripoSR, and NeRF stage of MeshLRM, SparseNeuS, etc. In contrast, approaches like GRM or the mesh stage of MeshLRM, which use pixel-aligned Gaussians and rasterization-based rendering respectively, capture textures much better due to their near-surface or surface representation. Similarly, our texture refinement uses UV representation with rasterization for rendering textures, ensuring textures exist only on the surface.
>
> While optimizing with respect to input images is a viable approach, it has issues:
> (a) Since the entire shape might not be visible in the input views, seams can appear between optimized visible and unoptimized non-visible areas, leading to incoherence and contrast mismatch.
> (b) Reconstructed objects are not perfect, leading to projection mismatches between the rendered images of the reconstructed object and the input images. This mismatch can provide incorrect supervision to the optimization objective, as the projections of imperfect 3D reconstructions may not align with the input images.
> (c) Time cost: Optimization takes a longer time.
>
> A learned approach like our texture refiner can combat both (a) and (b) since the training process learns to overcome these issues in a data-driven manner and can do so with a single feedforward pass, addressing the time cost (c).

---

> > ### Author Response · Authors · 2024-08-12
> >
> > With the reviewer-author discussion period nearing its end, we want to ensure that our responses have thoroughly addressed your primary concerns. We recognize the importance of this dialogue in resolving the issues raised by the reviewers and are committed to providing any further clarifications you might need. Please let us know if there are any additional questions or if more explanation is required.
> >
> > Warm regards,
> > Emu3D Authors

---

### Official Review · Reviewer_z8qF · 2024-07-15

**Soundness:** 4
**Presentation:** 4
**Contribution:** 3
**Rating:** 7
**Confidence:** 5

**Summary:**

The paper mainly works on text-to-3D with PBR materials. It follows the diffusion-based multiview generation and LRM-based reconstruction paradigm. It is a fast feed-forward solution for PBR generation: The diffusion model predicts both shaded and albedo, and the LRM predicts PBR via differentiable rendering with an efficient pixel-wise deferred shading loss. Moreover, it designs a UNet-based texture refiner to project input images to mesh, bringing sharp textures. Also, it adapts LightplaneLRM with the VolSDF renderer and SDF loss to enhance geometry. The generation only takes 30 seconds. It outperforms state-of-the-art feed-forward baselines and performs comparably to top optimization-based industry solutions.

**Strengths:**

1. Good performance.

    a. It compares with and outperforms some very recent SOTA approaches like InstantMesh and MeshLRM as well as some commercial products like Meshy and Luma Genie.

    b. It showcases that it can predict well on objects with mixed materials, which is very impressive. Its PBR prediction is SOTA performance. Also, it is not optimization-based and thus fast.

    c. Its texture refinement module is robust and preserves many details (Figure 4, 8), which helps win out in terms of visual appearance.
2. It conducts many quantitative and qualitative ablation studies to validate each component’s effectiveness.
3. The writing is very clear and easy to follow. Notations are used properly and are well explained. Each design choice is well-motivated and elaborated. It also includes a nice preliminary reading of the used BRDF model.

**Weaknesses:**

1. Image-to-3D may have a degraded material decomposition. Since the proposed approach does not predict albedo for input images, PBR prediction correctness will drop significantly as pointed out in the ablation study (Table 1, Fig. 3).
2. The diffusion model is inherently stochastic, which leads to variance in albedo prediction. How is this variance? How will this variance impact the material decomposition afterward? This point is not quantitatively evaluated.
3. The paper evaluates PSNR for metallic. It is great. But metallic is usually binary. A simple 2x2 confusion matrix on metallic prediction on single-material objects can show the performance more intuitively.
4. If input images are taken under non-uniform lighting conditions, I doubt if albedo can still be correctly decomposed.
5. Lack of failure case analysis. Following the previous point, I wonder if there are some typical failure cases for PBR prediction.

**Questions:**

- Fewer than 10% meshes are selected as high-quality samples to finetune Emu (L226). What is the filtering standard? In terms of appearance quality, material correctness, and composition complexity? Please elaborate more.
- L221: What dataset is used? Is it publicly available, like Objaverse? Is the dataset used under proper license?
- How is the text-to-image model (3-channel output) adapted to output 6 channels? Please provide more details on that.
- For results in Figure 7., if I understand correctly, they are deterministic? What is their PBR PSNR? (How cherry-picked are they?)
- May consider adding some qualitative comparisons to optimization-based PBR generation baselines like Richdreamer. Although they are time-costly and may not be as competitive as commercial solutions, such a comparison can strengthen the claim.
- In the Image-to-3D part (excluding texture refinement), Are albedo predictions only used as inputs to MLP?
- Some qualitative examples of Luma Genie 1.0, stage 2 vs the proposed approach (A.5). It would help to show where optimization helps to get an edge.
- 30-sec Runtime breakdown to each step.

**Limitations:**

Some suggestions are included in the Questions above.

---

> ### Author Rebuttal · Authors · 2024-08-06
>
> Thank you for detailed review and the helpful comments!
>
> 1. **Degraded PBR quality with only RGB shaded inputs** (weakness #1):
>
> Yes, indeed the PBR quality suffers when only RGB shaded inputs are provided (upper half Table 1) since the material decomposition is ambiguous. For this reason, the complete proposed approach for text-to-3D partially mitigates this problem by generating both the shaded and the albedo grid, which enhances the PBR quality (Fig. 4).
>
> 2. **Evaluation of variance in albedo and shaded RGB prediction** (weakness #2):
>
> Yes, for a given text prompt, the albedo and shaded RGB produced by the Emu model can vary due to the stochastic nature of diffusion. However, evaluating the effects quantitatively is challenging in the absence of the absolute ground truth. For evaluating the quality of outputs in the text-to-3D setup, we conducted a user study for user preferences using the fixed model seed, but evaluating the quality of material might be a bit more challenging for a non-professional user study participant.
>
> 3. **Confusion matrix for metalness** (weakness #3):
>
> It is true that, physically, materials are either metallic or dielectric, but computer graphics artists often use non-binary values of metalness, for example to represent materials that are a mixture of dielectrics and metals (e.g., certain rock conglomerates, transitions between oxidized/rusted and polished metal); in our own dataset, a non-trivial proportion of metallic assets have non-binary metalness. This justifies using PSNR or similar regression metric for metalness too.
>
> 4. **Non uniform lighting** (weakness #4):
>
> If the input views are captured with non-uniform lighting, it will indeed hamper PBR reconstruction performance. However, in our application, i.e., text-to-3D, lighting is controlled by fine-tuning the Emu grid generator on 3D models rendered with consistent lighting.This makes it biased toward generating images with consistent across views and with uniform lighting.
>
> 5. **Failure cases** (weakness #5):
>
> We will add a discussion and visualizations corresponding to failure cases in the final draft.
>
>
> ---
>
>
> ### **Answers to questions**
>
>
> 1. **Filtering standard for the 10K objects used for fine tuning Emu:**
>
> To filter the dataset to obtain the top 10K objects, we first remove those with too low aesthetic score. We then select the top 10K objects with the highest CLIP score (dot product between CLIP of the object render and CLIP of the caption). This number was empirically determined, as including more assets, i.e. including assets with a lower CLIP score, results in decreased text-image alignment. To control material quality, we only select assets with valid metallicness and roughness maps. Compositional quality is generally low, as a vast majority of assets in the dataset are single objects rather than scenes.
>
> 2. **Details about the dataset:**
>
> We use an internal proprietary dataset which is similar to Objaverse. The usage has been scrutinized by the professional legal counsel, which raised no concerns about copyrights. Copyright issues are why we did not use Objaverse.
>
> 3. **Adapting 3-channel Emu with 6-channel outputs:**
>
> We employ a text-to-image diffusion model pre-trained on billions of images annotated with text and expand its input and output channels by a factor of 2 to support simultaneous generation of the shaded appearance and albedo (Section A.6.1). This requires only the input and output layers to be adapted.
>
> 4. **Figure 7 results:**
>
> Yes, correct, the results are deterministic. Here, the task is sparse view reconstruction from 4 posed RGB images (without the albedo input), resulting in 3D meshes and their PBR materials. The results correspond to config G in Table 1, which shows the mean PBR PSNR on the whole test set. For the 6 images shown in the figure, the mean PSNR is 23.49 for albedo, 24.91 for metallicity, and 21.02 for roughness, with the standard deviation of (2.72, 3.86, 4.17) respectively. The object selection for the figure was intended to target objects with a non-uniform metalness or roughness (not just a single value) to make for an interesting visualization and highlight the model’s effectiveness.
>
> 5. **Comparison to Richdreamer:**
>
> Please check the PDF in the common response.
>
> 6. **How are albedo predictions used by the LRM:**
>
> In the vanilla Instant3D LRM, an image encoder encodes patchwise feature tokens of the image using a pre-trained DINO network (Caron et al. 2021). LightplaneLRM further extends this by adding Splatter layers on the DINO outputs in the subsequent image-to-triplane transformer. For accommodating additional albedo predictions, we add the encoded albedo patchwise features to the DINO network. This enables access to albedo as well as shaded image information to DiNO, and therefore subsequently to the image-to-triplane transformer and the splatting layers.
>
> We will add these details to the final draft.
>
> 7. **Qualitative Luma Genie 2.0 results:**
>
> Please check the PDF in the common response.
>
> 8. **Runtime breakdown:**
>
> Emu image generation is 20s, LRM reconstruction is 2s, meshification is 8s, and texture refinement < 1s.

---

> > ### Author Response · Authors · 2024-08-12
> >
> > With the reviewer-author discussion period nearing its end, we want to ensure that our responses have thoroughly addressed your primary concerns. We recognize the importance of this dialogue in resolving the issues raised by the reviewers and are committed to providing any further clarifications you might need. Please let us know if there are any additional questions or if more explanation is required.
> >
> > Warm regards,
> > Emu3D Authors

---

### Author Rebuttal · Authors · 2024-08-06

We thank the reviewers for their constructive and valuable feedback. We are pleased that all four reviewers found our presentation to be excellent, the soundness of our proposed approach to be good or excellent, and our contributions to be good.

We are glad the reviewers found our method to be “effective” (**cPQZ**), producing “efficient, high-quality results” (**Xevg**), and “favorable” (**Zgd2**), with “good performance” (**z8qF**, **cPQZ**). All reviewers appreciate the experiments, noting that the evaluation is “comprehensive“ (**cPQZ**) and “sufficient” (**Xevg**) with “many qualitative and quantitative studies” (**z8qF**, **Zgd2**), and “thorough ablations” (**Zgd2**) that “validate each component's effectiveness” (**z8qF**). The reviewers also found our design choices to be “well-motivated” (**z8qF**), “intuitive, and well-justified” (**Zgd2**). They further appreciated the writing quality as clear and easy to follow (**z8qF**, **cPQZ**) and well presented (**Xevg**).

Attached is a PDF with requested extra comparisons and visualizations, which will be included in the final draft. Due to limited space, we encourage the reviewers to zoom into the figures if necessary.

---

### Decision · Program_Chairs · 2024-09-25

**Decision:**

Accept (poster)

**Comment:**

This paper proposes a 2-stage feed-forward framework, named EmuLRM, for text-to-3D that predicts PBR materials to support realistic relighting. In the 1st stage, 4 views with both shaded and albedo channels are generated by a text-to-image diffusion model to asist the prediction of PBR materials in the 2nd stage. In the 2nd stage, the authors extend LightplaneLRM to predict SDF together with PBR materials. The authors also introduce deferred shading, a new scalable SDF-based render, and a direct SDF loss to improve training efficiency and scalability. Finally, they propose a texture refinement step operating in the UV space to improve the sharpness and details of the textures by leveraging the reference views through cross-attention.

This paper outperforms other feed-forward methods and produces sota results. Its results are also comparable to sota optimization based methods. Sufficient quantitative and qualitative results are presented to support and validate various design choices.

Overall, all reviewers are positive about this paper (accept, weak accept, weak accept, weak accept). The authors have addressed all the concerns rasied by the reviewers in their rebuttal. It is recommended to accept this paper.